# STEFLUX, a tool for investigating stratospheric intrusions: application to two WMO/GAW global stations

Davide Putero[1,2], Paolo Cristofanelli[1], Michael Sprenger[2], Bojan Škerlak[2], Laura Tositti[3], and Paolo Bonasoni[1]

[1]CNR–ISAC, National Research Council of Italy – Institute of Atmospheric Sciences and Climate, via Gobetti 101, 40129, Bologna, Italy
[2]IAC–ETH, Institute for Atmospheric and Climate Science – ETH Zurich, Universitätstrasse 16, 8092, Zurich, Switzerland
[3]Dept. Of Chemistry "G. Ciamician", Alma Mater Studiorum University of Bologna, Via Selmi 2, 40126, Bologna, Italy

*Correspondence to:* D. Putero (d.putero@isac.cnr.it)

**Abstract.** Stratospheric intrusions (SI) are a topic of ongoing research, especially because of their ability to change the oxidation capacity of the troposphere and their contribution to tropospheric ozone levels. In this work, a novel tool called STEFLUX (Stratosphere-to-Troposphere Exchange Flux) is presented, discussed and used to provide a first long-term investigation of SI over two global hot-spot regions for climate change and air pollution: the southern Himalayas and the central Mediterranean basin. The main purpose of STEFLUX is to obtain a fast-computing and reliable identification of the SI occurring at a specific location and during a specified time window. It relies on a compiled stratosphere-to-troposphere exchange (STE) climatology, which makes use of the ERA-Interim reanalysis dataset from the ECMWF, as well as a refined version of a well-established Lagrangian methodology. STEFLUX results are hereby compared to the SI observations (SIO) at two high-mountain WMO/GAW global stations in these climate hot-spots, i.e., the Nepal Climate Observatory-Pyramid (NCO-P, 5079 m a.s.l.) and Mt. Cimone (2165 m a.s.l.), which are often affected by SI events. Compared to the observational datasets at the two specific measurement sites, STEFLUX is able to detect SI on a regional scale. Furthermore, it has the advantage of retaining additional information concerning the pathway of stratospheric-affected air-masses, such as the location of tropopause crossing and other meteorological parameters along the trajectories. However, STEFLUX neglects mixing and dilution that air-masses undergo along their transport within the troposphere. Therefore, the regional-scale STEFLUX events cannot be expected to perfectly reproduce the point measurements at NCO-P and Mt. Cimone, which are also affected by small-scale (orographic) circulations. Still, the SI seasonal variability according to SIO and STEFLUX agree fairly well. By exploiting the fact that the ERA-Interim reanalysis extends back to 1979, the long-term climatology of SI at NCO-P and Mt. Cimone is also assessed in this work. The analysis of the 35-year record at both stations denies the existence of any significant trend in the SI frequency, except for winter seasons at NCO-P. Furthermore, for the first time, by using the STEFLUX outputs, we investigate the potential impact of specific climate factors (i.e. ENSO, QBO and solar activity) on SI frequency variability over the Mediterranean basin and the Himalayas.

# 1 Introduction

Stratosphere-to-troposphere exchange (STE) represents one of the natural processes that have substantial impacts on meteorology and atmospheric chemistry, and is an important aspect of climate change (Appenzeller and Davies, 1992; Holton et al., 1995; Stohl et al., 2003; Stevenson et al., 2006). The definition of STE encompasses a two-way air-mass transport:
the downward transport from the stratosphere to the troposphere (STT) and the upward transport from the troposphere to the stratosphere (TST). A specific type of STT is called stratospheric intrusion (SI), which we hereby define as the downward transport of stratospheric air-masses relatively deep into the troposphere (as done in Cristofanelli et al., 2006). SI are capable of changing the oxidation capacity of the troposphere (Gauss et al., 2003) and their contribution to the ozone ($O_3$) levels in the troposphere has been estimated to be as large as the net photochemical production (Roelofs et al., 1997), although models still
show large uncertainties in the estimates (e.g., Stevenson et al., 2006; Young et al., 2013). As pointed out in many studies (e.g., Reed, 1955; Appenzeller and Davies, 1992; Lamarque and Hess, 1994; Holton et al., 1995; Appenzeller et al., 1996; Stohl et al., 2003; Cooper et al., 2005; Sprenger et al., 2007) SI can be caused by different mechanisms and are typically associated to distinct synoptic- and meso-scale features: tropopause folds and cutoff lows, subtropical jet streams and streaks, potential vorticity (PV) streamers, upper-level fronts and anticyclonic areas.

High mountain stations are appropriate sites for investigating the transport of stratospheric air-masses into the troposphere, because stratospheric air-masses can already be identified at mid-tropospheric levels. Furthermore, they are less influenced by polluted air-masses due to local or regional anthropogenic emissions (Stohl et al., 2000), which makes the SI detection more straightforward. Several studies have been carried out in the past, to assess the influence of SI at high-altitude remote sites, which also represent ideal locations for studying the background conditions of the troposphere (e.g., Stohl et al., 2000;
Cristofanelli et al., 2006; Ordóñez et al., 2007; Cristofanelli et al., 2010; Trickl et al., 2010; Lin et al., 2012). Usually, stratospheric influence is detected at a measurement site by analyzing the variability of in situ "stratospheric" observations (e.g., relative humidity, $^7Be$, $^{10}Be$, $O_3$, atmospheric pressure variability) and profiling datasets (radio/ozone-sondes), coupled with the analysis of satellite (e.g., total column of ozone) and various kinds of numerical weather prediction (NWP) model products fields. Many different methods, as thoroughly reviewed in Stohl et al. (2003), are based on this combined approach. Stohl et al.
(2000) deployed a detection algorithm based on the in situ variation of experimental data and simulations with a passive stratospheric tracer. Similarly, other studies analyzed STE by coupling experimental data and back-trajectories (e.g., Cristofanelli et al., 2006, 2010; Trickl et al., 2010). Usually, specific threshold values are applied to in situ tracers' variability to detect the presence of air-masses with stratospheric "fingerprints". Also trajectory and dispersion models are extensively used to detect the occurrence of STE. For example, Cui et al. (2009) used the particle dispersion model FLEXPART (Stohl et al., 2005) and
the trajectory model LAGRANTO (Wernli and Davies, 1997) to identify stratospheric transport at the high-altitude Alpine site Jungfraujoch (Switzerland), while Tarasova et al. (2009) deployed 3D air-mass back-trajectories to trace the atmospheric transport at two high mountain measurement sites over the Alps and Caucasus. As pointed out by Bourqui (2006), trajectory-based approaches can provide a lower-bound estimate for STE flux, while dispersion models can provide slightly larger estimates. Typically, when used to detect STE at specific locations at the Earth's surface, all of these "observations-based" methodologies

vary among different measurement sites, with respect to the number and types of stratospheric tracers available/considered, threshold values adopted, and often require a lot of time-consuming implementation to work. Moreover, it should be argued that none of the most diffused tracers have a "pure" stratospheric origin; for example, $^7$Be and $O_3$ are affected by significant tropospheric sources. Furthermore, the compilation of proper long-term climatologies is very often hindered by the lack of

long-term observations of "stratospheric" tracers.

In this work we present a novel tool, which aims at objectively identifying SI reaching a "target" geographical region and during a specific time window. The tool, called STEFLUX (Stratosphere-to-Troposphere Exchange Flux), is a relatively fast-computing algorithm which makes use of the pre-computed trajectories composing the STE climatology by Škerlak et al. (2014). This climatology is available from 1979 and continuously updated. The Lagrangian approach, on which it is based,

has been extensively used in previous studies (e.g., Wernli and Bourqui, 2002; Sprenger and Wernli, 2003; Bourqui, 2006; Sprenger et al., 2007; Škerlak et al., 2014), and has been confirmed to effectively identify SI events and to reproduce several of their related aspects. Its computational speed and user-friendly approach (it is sufficient to specify only a few parameters to work) make it suitable for obtaining a quick and reliable estimate of the SI occurred at a specific place over the desired time window (including long periods which would otherwise require a lot of time-consuming calculations). A potential use

of STEFLUX is to identify the SI occurrence in locations where a detection based on observational data is not available. To evaluate the STEFLUX skills in identifying the SI events, we hereby compare its outputs with the SI identification based on observations at two high-altitude World Meteorological Organization/Global Atmosphere Watch (WMO/GAW) global stations in Asia (Nepal) and Europe (Italy). Then, we use STEFLUX to provide a first investigation on the long-term (i.e., 1979–2013) variability of SI occurrence at these measurement sites, that are indeed representative of the lower troposphere of two hot-spot

regions for climate change and anthropogenic impacts on climate (Monks et al., 2009): the central Mediterranean basin and the southern Himalayas. In particular, we provide a first assessment of possible impact of large-scale climate processess (i.e. ENSO, QBO, solar activity) in modulating the long-term SI variability.

The paper is structured as follows: in Sect. 2 we define the selection methodologies that were used for identifying SI events at the two measurement sites; in Sect. 3 we describe in detail the STEFLUX tool, along with a case study to show a potential

application. STEFLUX time series are then compared to the in situ measurements in Sect. 4, followed by a critical discussion about the benefits and restrictions of the tool. Furthermore, trends and periodicities of the long-term SI time series at NCO-P and Mt. Cimone are assessed. Finally, Sect. 5 summarizes the main results of the study.

## 2  Experimental datasets

Datasets of daily SI occurrences are available at two high-altitude WMO/GAW global stations, i.e., the Nepal Climate Observatory-

Pyramid (NCO-P, 5079 m a.s.l., Nepal) and Mt. Cimone (2165 m a.s.l., Italy) since 2006 and 1998, respectively. In this section, a brief description of the two measurement sites is provided, together with the description of the methodology used to detect SI events based on the analysis of in situ stratospheric tracers' variability (coupled with additional model data). Hereinafter,

these datasets will be referred to as "Stratospheric Intrusions Observations (SIO)". Technical details on the different parameters considered are given in the Supplementary Material and in the papers by Cristofanelli et al. (2006, 2010).

NCO-P (27.95° N, 86.82° E) is located in the southern Himalayas, near the base camp of Mt. Everest, in the Khumbu Valley, Nepal. This station is far away from anthropogenic sources, thus it can be considered representative of the background conditions of the high Himalayas and the free troposphere (especially during night-time). Further details on the measurement site and on the instrumental setup are given in Cristofanelli et al. (2010). To account for days likely affected by SI events at NCO-P, a specifically designed statistical methodology was applied to the time series of observed and modeled variables. The parameters used consisted of in situ measurements ($O_3$, atmospheric pressure – P and relative humidity – RH), satellite observations (total column of $O_3$ – TCO, as retrieved by the OMI – Ozone Monitoring Instrument) and NWP-based back-trajectories (by using LAGRANTO, see Sprenger and Wernli, 2015; Wernli and Davies, 1997). The methodology is composed of four different criteria; at least one must be satisfied to identify a day as likely influenced by SI:

1. significant variations of daily P values and presence of back-trajectories with values of PV > 1.6 pvu;

2. significant daily TCO increases and presence of back-trajectories with values of PV > 1.6 pvu;

3. significant variations of daily P values and significant TCO daily increases;

4. presence of RH values lower than 60% and significant negative correlation O3-RH and daily $O_3$ maximum higher than the seasonal value and significant variation of daily P, PV or TCO values (this last criterion was introduced for taking into account the possible role of downward valley winds in transporting air-masses from aloft).

The significant variations are obtained as follows: first, a three-time repeated iteration of a 21-days running mean (the so-called Kolmogorov-Zurbenko filter, see Sebald et al., 2000) is applied to the daily average time series, and residuals are calculated by subtracting these values from the daily averages; then, it is checked whether residuals exceed the upper or lower endpoints of the 95% confidence interval of the residuals distribution over the whole period. For this work, the period of study considered for NCO-P spans from March 2006 to December 2013.

Mt. Cimone (44.19° N, 10.70° E) is the highest peak of the Italian northern Apennines. The observations carried out at this sampling site can be considered representative of the free tropospheric conditions for most of the year, while during warm periods the station can be affected by thermal and convective transport of planetary boundary layer (PBL) air. Other details about Mt. Cimone and the instrumental setup can be found in Cristofanelli et al. (2015) and references therein. Similarly to NCO-P, in situ measurements of Beryllium-7 ([7]Be) and RH have been considered, as well as satellite observations (TCO, as deduced by TOMS–Total Ozone Mapping Spectrometer–and OMI overpass data) and PV of air-masses reaching the sampling site (by the analysis of 7-day FLEXTRA back-trajectories, Stohl et al., 1995). The statistical method for identifying SI days is based on the following four criteria:

1. significant daily TCO increases and presence of back-trajectories with values of PV > 1.6 pvu;

2. significant daily [7]Be increases and presence of back-trajectories with values of PV > 1.6 pvu;

3. presence of RH values lower than 40% and presence of back-trajectories with values of PV > 1.6 pvu;

4. presence of RH values lower than 40% and significant TCO daily value increases.

Again, the significant variations are defined in the same way as done for NCO-P, and at least one criterion must be valid for tagging the selected day as influenced by SI. The period of study for Mt. Cimone spans from January 1998 to December 2010.

SIO at these two measurement sites provide a unique opportunity to test the capacity of STEFLUX in reproducing the main features of SI occurrences (frequency, seasonality, long-term variations) at two locations representative of the Northern Hemisphere midlatitudes and subtropics.

## 3   The STEFLUX tool

### 3.1   Description of the tool

The main purpose of STEFLUX is to obtain a fast-computing and reliable estimation of SI occurring at a specific location. The database used as input relies on the trajectories from the STE climatology presented in Škerlak et al. (2014), which makes use of the ERA-Interim reanalysis dataset from the ECMWF (Dee et al., 2011), as well as a refined version of a well-established Lagrangian methodology (Wernli and Bourqui, 2002), to calculate mass and ozone fluxes across the tropopause and several pressure surfaces. For further details, please refer to Škerlak et al. (2014).

STEFLUX detects the air parcels originating in the stratosphere and entering a tropospheric 3D target box during a specific time window. For this reason, several parameters need to be defined for the STEFLUX tool to work: (i) the time period for which the analysis should be carried out, and (ii) the target box by means of its longitude and latitude boundaries and by its vertical extension from the surface up to the top boundary (defined as a pressure level, in hPa). On request, the PBL height can be used as top boundary of the target box: this option takes into account the ERA-Interim PBL height, which is a parameter

also available along each STE trajectory. In addition to this, another optional parameter allows the temporal resolution for the STE trajectories to be increased from its default value (6 h) up to 1 h. STEFLUX produces several output files, which enclose: (i) the trajectory positions and timing found within the box, (ii) the first box crossing positions and timing for each trajectory, (iii) the tropopause crossing position and timing for each trajectory, (iv) the complete list of the trajectories that have crossed the box.

### 3.2   Illustrative case study

To present an application of STEFLUX and its output files, a SI case in January 2007 is discussed. The period of study coincides with a case study discussed in Bracci et al. (2012), where the SI event is strongly related to the subtropical jet stream. We defined a box with horizontal extension 85–88° E and 26–29° N, centered at NCO-P. The top boundary of the box was taken at 550 hPa (the average pressure at the station), while the STEFLUX tool was run for the time period 9–25 January 2007. All STE

trajectories from the climatology by Škerlak et al. (2014), introduced in Sect. 3.1, were analyzed and their crossings of the target box boundaries were determined.

Figure 1a shows as a time series the number of the daily crossings (derived from the STE trajectories) in the box, according to STEFLUX. Additionally, the daily averaged values for $O_3$ and RH at NCO-P are shown, and each day is also marked as selected or not by the SIO methodology, according to the criteria introduced in Sect. 2. The study period was characterized by the presence of clean and dry (daily RH average always below 40%) air-masses; moreover, the double-jet structure of wind speed at 250 hPa (contour lines in Fig. 1b) indicated the presence of the subtropical jet stream over South Asia and the Himalayas. SIO methodology identified a likely SI event, spanning from 13 to 17 January 2007 (the missing record of $O_3$ during 15 January was responsible for the gap in the SIO time series). This was confirmed by STEFLUX: from 12 to 24 January 2007, several STE trajectories passed through the target box; at its peak, 25 daily crossings were counted. This time series is directly based on two STEFLUX output files, which list the first and the entire trajectory positions found within the target box. In addition to the crossing time, the list of variables includes the position (longitude/latitude/pressure) and several meteorological parameters (e.g., potential temperature, specific and relative humidity, PBL height) at each point within the target box.

The first positions of the air parcels after entering the target box are marked in Fig. 1b as blue dots; additionally, the positions where the air parcels crossed the dynamical tropopause are shown as green dots. Interestingly, these crossings are mainly clustered into a region over North Africa, which is identified as a preferred region for tropopause crossing in previous studies (Sprenger and Wernli, 2003; Škerlak et al., 2014), and also in this paper (see Sect. 4.2.1). The exact times and positions of the tropopause crossings are saved by STEFLUX in a third output file, together with atmospheric pressure, potential temperature and $O_3$ concentration. The output files from STEFLUX allow the history of the STE air parcels to be studied along their way from the stratosphere to the target box. As an example, Fig. 1b reports all the trajectories from the tropopause crossings (green dots) to the target box (blue dots). These trajectories are colored according to their PV value, i.e., points below 2 pvu (magenta) and points greater than 2 pvu (black). Fig. 1c shows the time-height evolution of the trajectories, where time is given relative to the arrival time in the target box. Additionally, the top boundary height of the target box (550 hPa) is reported in the figure (red horizontal line). It is discernible that most of air parcels slowly descended until 72–48 h before they reached the box. Previously, they were characterized by PV values above 2 pvu (black line), i.e., they still followed the stratospheric circulation steered by the subtropical jet stream (as also discernible from the contour lines of wind speed at 250 hPa in Fig. 1b, which mark the double-jet structure). A rapid descent then set in before their arrival; hence, the PV falls below 2 pvu, indicating that the air parcels crossed the dynamical tropopause.

## 4 Results

### 4.1 STEFLUX vs SIO

In this section, the SI occurrences from STEFLUX are compared to the ones from SIO at the two WMO/GAW global stations (see Sect. 2). For both stations, STEFLUX was run by setting a target box with a horizontal extension of $3° \times 3°$ around the measurement site, after performing a sensitivity test on this parameter (not shown). Vertically, the box extended up to 550 hPa for NCO-P and 790 hPa for Mt. Cimone, respectively, corresponding to the average pressure level recorded at each station

throughout the year. The selected time periods were the same as in Sect. 2 (i.e., March 2006–December 2013 for NCO-P and January 1998–December 2010 for Mt. Cimone), but the temporal resolution for the STE trajectories was increased to 1 hour (see Table 1 listing all of the input parameters).

The aim of this part is twofold: first, we would like to see how STEFLUX compares to SIO (in Sect. 4.1.1 and 4.1.2). This comparison, which turns out to be not a perfect match, will lead to a critical discussion of what can and cannot be expected from STEFLUX. Hence, it is paramount to understand that both STEFLUX and SIO have complementary strong and weak points in identifying SI and thus may not be exactly compared one-to-one (as discussed in Sect. 4.1.3).

### 4.1.1   Seasonal comparison and inter-annual variability

The seasonal frequency (in %) of SI days, derived from measurements (SIO), is presented in Fig. 2 as a red line. Additionally, the seasonally averaged percentage of available data from each criterion (hereinafter referred to as "criteria coverage") is also reported in the plot (grey bars). Note that the season definition slightly differs for the two sites: at NCO-P it consists of a dry (winter – DJF), a wet (monsoon – JJAS) and two transition seasons (pre-monsoon – MAM and post-monsoon – ON) (see Bonasoni et al., 2010), while for Mt. Cimone the "classic" Northern Hemispheric definition is chosen (winter – DJF, spring – MAM, summer – JJA, and autumn – SON). A clear seasonality characterized the SIO frequency at both stations, as highlighted by the seasonally averaged SI frequencies obtained. For NCO-P (Fig. 3a), a maximum was discernible in winter and a minimum during the monsoon season, while for Mt. Cimone (Fig. 3b) high SI values were found in winter and spring and a minimum in summer. We computed the same time series of seasonal frequencies using STEFLUX (blue lines in Fig. 2), with a threshold of at least 2 box crossings per day, in order to retain robust information only and to discharge "erratic" events. It showed a clear average seasonality at both stations (Fig. 3c,d), comparable and consistent with that from SIO, especially for NCO-P. At Mt. Cimone, the average annual variation for STEFLUX was more pronounced than the one derived from SIO. This was due to an overestimation of the STEFLUX average frequency for November–January and to an underestimation for June–July. However, the observed seasonality is in line with previous works (e.g., Trickl et al., 2010; Škerlak et al., 2014).

Although the seasonality was a feature well captured by STEFLUX, the inter-annual variability was less clearly identifiable. Concerning NCO-P (Fig. 2a), the correlation between the two seasonal time series was rather high (Pearson's $r = 0.7$), but for specific years STEFLUX and SIO results evidently differed in the amplitude and timing of the annual peaks. In particular, SIO showed much higher SI frequency than STEFLUX for post-monsoons during 2010–2012. When comparing seasons individually, the correlation coefficient was satisfying for pre-monsoon, monsoon and post-monsoon ($r = 0.5$ on average). For winter, the two time series are even anti-correlated ($r = -0.4$). This can be attributed to a significant decrease of SI detection by SIO during winter for the years 2009–2010. Moving to Mt. Cimone (Fig. 2b), the correlation between the SIO and STEFLUX time series was still high ($r = 0.7$). Also the individual comparison of the seasons gave satisfying results ($r$ varied between 0.4 and 0.5). It has to be noted the evident decrease in SI detection coverage at Mt. Cimone during the period 2006–2011, related to a lower availability of [7]Be observations at this sampling site (measurements were stopped in 2012, see Tositti et al., 2014). To investigate whether the low SIO coverage hindered the comparison with STEFLUX, the same analysis was also performed by

limiting the Mt. Cimone dataset to the period 1998–2004, i.e., when the criteria coverage was greater than 90% for all of the seasons. However, the results did not significantly differ (see Table S1 in the Supplementary Material).

### 4.1.2 Event-based comparison

In this section, we extend the comparison to a higher temporal resolution, i.e., by considering single SI events. More specifically, in this study, a SI event was defined (for both STEFLUX and SIO) as the aggregation of contiguous SI days. Furthermore, cases in which two distinct SI events were separated by a single no-SI day were treated like a single event covering the entire period. Generally, an event-based comparison between modelled and observed SI events and experimental detection is a very challenging task, as pointed out by previous investigations, concerning the transport and mixing of stratospheric air deep into the troposphere (e.g., Meloen et al., 2003; Cui et al., 2009; Bracci et al., 2012).

For NCO-P, based upon the SIO criteria, a total of 203 SI events (361 days influenced by SI, representing 13% of the period) were identified, with duration ranging from 1 to 14 days, and average length of 1.9 days. On the other hand, STEFLUX identified 155 SI events (376 days, 13%), with duration ranging from 1 to 10 days (average length: 2.6 days). At Mt. Cimone, 299 SI events (433 days, 9%) were identified by the SIO methodology (with duration ranging from 1 to 8 days, and average length of 1.6 days), while STEFLUX yielded 237 SI events (491 days, 10%) that lasted from 1 to 10 days (with an average length of 2.2 days).

To assess the STEFLUX performance, the approach presented in Cui et al. (2009) was followed. First, all SI events as retrieved by SIO were considered, and then it was checked whether at least 50% of the duration of each SIO event was confirmed by STEFLUX. If this was the case, STEFLUX was considered able to capture the selected SIO event. Hereinafter, we will refer to this comparison as "SIO vs STEFLUX". Vice-versa, the "STEFLUX vs SIO" comparison checked if a SI event (as defined by STEFLUX) was confirmed by the SIO dataset.

As an overview of the results of this comparison, we computed contingency tables (Table 2). In these 2×2 tables, each entry encloses a list of SI or no-SI events, as defined by the considered methodology (STEFLUX and SIO). From the contingency tables it is possible to evaluate several skill scores, which are useful to measure the skill of one method in identifying SI events compared with the other one. The accuracy (ACC), false alarm ratio (FAR), and probability of false detection (POFD) skill scores are defined, according to Thornes and Stephenson (2001) and Wilks (2006), as:

$$ACC = \frac{A+D}{A+B+C+D} \tag{1}$$

$$FAR = \frac{B}{A+B} \tag{2}$$

$$POFD = \frac{B}{B+D} \tag{3}$$

where, for each contingency table, A represents the number of SI events selected by both methodologies (STEFLUX and SIO); B represents the number of events selected as SI by the first methodology but as no-SI by the second one; C represents the

number of events selected as no-SI by the first methodology but as SI by the second one; and D represents the number of no-SI events selected by both methodologies. All four contingency tables give identical values of ACC (0.58) and POFD (0.45), while FAR varies between 0.73 and 0.78. The rather high FAR values and low POFD values can be partially explained by considering that the occurrence of SI is a relatively "unlikely" event with respect to the occurrence of no-SI. Also for taking into account this point, we considered an additional parameter (i.e., the Odds Ratio Skill Score, ORSS, see Thornes and Stephenson, 2001), which is not influenced by the marginal totals (i.e., $A+C$ and $B+D$). This parameter is defined as:

$$ORSS = \frac{A \times D - B \times C}{A \times D + B \times C} \tag{4}$$

The ORSS varies between -1 and +1, where a score of 1 represents perfect skill and a score of 0 indicates no skill; negative values imply that values of one series are opposite to what observed by the other one. Also reported in each table is the minimum ORSS required to have real skill at the 99% confidence level (see Thornes and Stephenson, 2001). All of our combinations indicate that the agreement between the methodologies is not due to chance (i.e., is statistically significant). Mt. Cimone showed higher scores than NCO-P. This can be explained by the location of NCO-P: it is placed at the bottom of a narrow valley (see Bonasoni et al., 2010) and therefore subgrid-scale processes (e.g., PBL entrainment and thermally driven valley winds, not reproduced by the trajectories analyzed by STEFLUX) play an important role in transporting stratospheric air-masses from the free troposphere to the surface (Cristofanelli et al., 2010).

Table 3a focuses on the "SIO vs STEFLUX" comparison, as a function of the length of the different events. STEFLUX captured one quarter of the measured events for NCO-P and Mt. Cimone. The highest agreement was found for 2-day events at NCO-P (42%), and for 3-day events at Mt. Cimone (39%). In particular, all the longest events were confirmed by STEFLUX at NCO-P (9- and 14-day long events), while at Mt. Cimone only 2 of 4 events longer than 7 days were captured. Finally, the "STEFLUX vs SIO" approach is correspondingly assessed in Table 3b. One quarter of the SI events observed by STEFLUX were confirmed by SIO at NCO-P and Mt. Cimone (25% and 22%, respectively). Again, the maximum agreement was found for events that lasted 2 days, while the minimum agreement was assessed for 1-day long events.

### 4.1.3 STEFLUX and SIO: strong and weak points

Several possible reasons can explain the mismatch between the SIO and STEFLUX time series. For instance, STEFLUX is not fully able to capture subgrid-scale processes (like convection, turbulent diffusion and mixing) along the path from the stratosphere to the target region. This deficiency becomes particularly pronounced over mountainous measurement sites, mostly because of the complex topography and the associated small-scale thermally and dynamically driven circulations that characterize the area. As shown in Bracci et al. (2012), it is common that stratospheric air-masses reach the upper tropospheric layers over NCO-P, without directly arriving at the station altitude. Then, the air is trapped and mixed within the PBL and thus brought to the measurement station. It was for this reason that a specific criterion was introduced in the SIO detection methodology at NCO-P (see criterion 4 in Sect. 2). It is worth noting that for NCO-P the largest bias between STEFLUX and SIO was observed when this criterion dominated the detection of SI (post-monsoons 2010 and 2011). Since the mixing processes might take several hours, this could be the reason of the lower agreement between SIO and STEFLUX at NCO-P.

In case of long travel times from the tropopause to the target region, we expect a stronger impact of mixing and dilution processes on the air-mass properties. Hence, when a SI actually affects a specific region, the SIO criteria might not be able to detect it, because mixing and dilution with tropospheric air-masses could lower stratospheric tracers concentrations below the thresholds used for detection. Then, we computed the travel times (hereinafter called $\Delta t$, expressed in hours) between the tropopause crossing and the first box crossing for each SI event. To evaluate the possible dependence from the travel time as a function of seasons, we sorted $\Delta t$ into five categories (from 0 to 120 h, divided into 24 h intervals), and then we calculated the seasonal occurrence and the annual variation of each category (Fig. S1 and S2 in the Supplementary Material). The maximum value for $\Delta t$ was chosen according to the typical lifetime values for a stratospheric intrusion into the troposphere (see Stohl et al., 2000; Bourqui and Trépanier, 2010; Trickl et al., 2014, 2016). On average, one third (32% and 30% for NCO-P and Mt. Cimone, respectively) of the SI events identified by STEFLUX presented maximum travel times (96 h $\leq \Delta t <$ 120 h). Furthermore, SI events characterized by relatively long (i.e., $\Delta t \geq$ 72 h) travel times usually dominated all the seasons. This suggests that a significant impact of dilution/turbulence small scale processes along stratospheric air-mass transport is likely and might explain part of the mismatch between STEFLUX and SIO. This hypothesis was further confirmed by analyzing the events seen by STEFLUX, but not confirmed by SIO, as a function of $\Delta t$: most of them (86% and 88% for NCO-P and Mt. Cimone, respectively) were characterized by medium/long travel times (i.e., $\Delta t \geq$ 48 h).

A further point of discrepancy between STEFLUX and SIO results is related to the "overpasses" phenomenon, i.e., air-masses that overpass the station at altitudes high enough that there is no indication in the measurements record (but might be observed by STEFLUX). Indeed, during a study conducted at the Zugspitze (Germany, 2962 m a.s.l.), Trickl et al. (2010) showed that overpasses explained nearly the 20% of occurrences that were not identified by the observations. Furthermore, for NCO-P, it should be considered that the station is located in a narrow valley. Thus, it is conceivable that, during the transport within the valley, $O_3$ (one of the stratospheric tracers considered by SIO) experiences deposition phenomena, thus decreasing the actual concentration that the stratospheric air-mass would have in the free troposphere (see, e.g., Furger et al., 2000; Wotawa and Kromp-Kolb, 2000).

In summary, although correctly depicting the typical seasonal variability of SI frequency at NCO-P and Mt. Cimone, the STEFLUX and SIO time series differ for several reasons. These differences point out that the complete approach to study and assess SI is to deploy together modeling tools and observations, because they are complementary and address together several scientific questions. Especially, in situ observations have the advantage of capturing short and transient SI events associated to transport processes occurring at subgrid scales, while STEFLUX has the advantage of detecting the arrival of stratospheric-affected air-masses, irrespective of the degree of mixing and dilution along the transport within the troposphere.

## 4.2 Long-term evaluation of SI occurrences at the two measurement sites

### 4.2.1 SI events climatology

In this section we present a climatology of SI events, as defined in Sect. 4.1.2, for the whole STEFLUX dataset (back to 1979, i.e., when the trajectories from the ERA-Interim reanalysis were first available). This allowed us to cover a 35-year period

(1979–2013) of monthly SI frequency values. In total, we obtained 673 SI events at NCO-P (representing 13% of the period), with an average length of 2.6 days. For Mt. Cimone, the number of SI events was lower (592, representing 9% of the period), as well as the average duration of an event, i.e., 2.1 days. The percentage of events with length equal to or less than 4 days was very high for both stations (86% and 93% considering all data, for NCO-P and Mt. Cimone, respectively), with peaks up to 98% and 100% in the summer season. On the other hand, longer events were observed during winter at NCO-P and were more equally spread throughout the rest of the year at Mt. Cimone. The seasonal cycle was also computed for these two longer time series (see Fig. S3 in the Supplementary Material); the seasonality considering all monthly data was confirmed and comparable to that obtained in Fig. 3, for both measurement sites.

An important aspect of SI is where and when the SI trajectories actually crossed the tropopause. First, the location of the crossing is useful to determine the $O_3$ concentration of the air parcels at the start of their tropospheric path towards the target region. Second, as mentioned in Sect. 4.1.3, we can expect that a longer time since the tropopause crossing goes along with enhanced dilution until its arrival in the target region, although keeping in mind that the diluting processes along the path can be highly transient and nonlinear in time. As introduced in Sect. 3.2, STEFLUX allows the position and time of the tropopause crossings to be analyzed. This allowed us to compute a tropopause crossing density plot for each measurement site over the entire 35-year period, as presented in Fig. 4. For NCO-P (Fig. 4a), the tropopause crossings associated with SI events are predominantly found over two areas, i.e., Central Asia and Northeast Africa, close to the Mediterranean Sea. On the other hand, no prevalent locations characterize the tropopause crossing for SI events at Mt. Cimone (Fig. 4b), where the crossings are spread over a large area extending from North America to the northern Europe. Considering seasons separately, a small cluster emerges south of Greenland during winter, while the other seasons still maintain the crossings spread over a larger area (not shown). These results agree with previous climatological studies (Sprenger and Wernli, 2003; Škerlak et al., 2014), which indicated that the tropopause crossing predominantly occurs over the Atlantic and Pacific storm track regions (in winter, spring and autumn), over the Mediterranean (in winter and spring) and over southeastern Europe and Central Asia (in summer). The tropopause crossing locations were then categorized according to $\Delta t$ (defined in Sect. 4.1.3), see Fig. S4 and S5 in the Supplementary Material. For NCO-P, the Central Asia zone of tropopause crossing was pronounced for all $\Delta t$ categories, up to 96 h, while the Northeast Africa cluster was absent for events with low $\Delta t$ and clearly discernible for events with 48 h $\leq \Delta t <$ 72 h, stressing the importance, for the southern Himalayas, of the fast transport of stratospheric air-masses embedded within the subtropical jet stream (Bracci et al., 2012). As stated above, the fact that SI events at Mt. Cimone showed no preferred tropopause crossing locations was confirmed for all the different $\Delta t$ categories.

### 4.2.2 SI frequency long-term trends and variability

To detect potential trends in the SI frequencies, we adopted the same STEFLUX climatological datasets presented in Sect. 4.2.1. Trends were calculated by using the Theil-Sen (Theil, 1950; Sen, 1968) regression method implemented in the Openair software (Carslaw and Ropkins, 2012), after having deseasonalized the time series. No significant trends in the SI frequency were discernible for the stations, which both showed only a weak increase of 0.03 % $yr^{-1}$. In addition to this estimation, we repeated the trend analysis based on seasonal SI frequencies. The only significant ($p < 0.1$) positive trend was found for winter

at NCO-P (0.18 % yr$^{-1}$). The lack of an overall trend in SI events was in line with previous findings, such as Sprenger and Wernli (2003) and Škerlak et al. (2014).

The long-term variability of SI frequency at NCO-P and Mt. Cimone was further analyzed with respect to potential oscillations and periodicities. To this aim, we applied the Complete Ensemble Empirical Mode Decomposition with Adaptive Noise method (CEEMDAN, Torres et al., 2011). This technique is an improved version of the original EMD method (Huang et al., 1998) based on the Hilbert-Huang transform and practical for non-linear and non-stationary time series. It aims at subtracting several components (i.e., the so-called Intrinsic Mode Functions, IMFs) from the original signal, each of which explains a different cyclic variation, and a residual, which represents the overall trend in the original time series. The method has been recently used in atmospheric and climatic studies (e.g., Coughlin and Tung, 2004; Xu et al., 2016), but none of these regarded trends in SI yet. For both sites, the time series could be decomposed into 7 IMFs, with very different time scales (Fig. 5 for NCO-P and Fig. 6 for Mt. Cimone). Since we were particularly interested in long-term variations, we neglected high-frequency oscillations with characteristic periods shorter than one year. Apart from an evident seasonal cycle (Fig. 5b), the time series at NCO-P presents an IMF with a clear period of 28 months (IMF5, Fig. 5c) that is weakly anti-correlated ($r$ = -0.3) to the Quasi-Biennial Oscillation (QBO); the anti-correlation is maximized during post-monsoon and winter seasons ($r$ = -0.5 and $r$ = -0.4, respectively). In this work we adopted the QBO index (blue line in Fig. 5c) for comparison, archived by the Free University of Berlin (http://www.geo.fu-berlin.de/met/ag/strat/produkte/qbo/qbo.dat). It refers to the monthly equatorial zonal wind at 50 hPa. Signals relating STE and QBO were found by Hsu and Prather (2009), who indicated that 20% of the inter-annual STE variance in the Northern Hemisphere can be explained by the QBO. More generally, the mechanisms for which QBO affects the STE variability are both the direct modulation of the circulation through thermal wind balance, and the impact on the strength of the overturning circulation by altering the propagation and dissipation of planetary-scale waves (Tung and Yang, 1994; Kinnersley and Tung, 1999; Neu et al., 2014). IMF6 (Fig. 5d) exhibits two peaks in the power spectrum, corresponding to periods of 3.5 and 5.8 years (not shown), potentially indicating an influence from the El-Niño/Southern Oscillation (ENSO). In fact, ENSO has been found to have an impact on the STE variability via the induced anomalous strong convective activity in the tropics (James et al., 2003). Moreover, a strong correlation between STE and ENSO was found by Zeng and Pyle (2005) and Voulgarakis et al. (2011), with the total modeled STE maximized during El Niño and minimized during La Niña years. The link is probably caused by modulations of the subtropical jet. In our work, IMF6 presents some periods of inverse variability with respect to the Multivariate ENSO Index (MEI, Wolter and Timlin, 1993), included as the red line in Fig. 5d. Similar relations were also reported in Neu et al. (2014), with the STE flux increased during El Niño/easterly-shear QBO, because of the strengthening of the stratospheric overturning circulation and the intensified transport of air from the ozone maximum poleward and downward to midlatitudes. Conversely, La Niña/westerly QBO phases are associated with a weakening of the circulation and hence reduced STE flux. The last IMF (IMF7, Fig. 5e) shows a period of nearly 10 years, possibly related to the solar cycle. The time series of the 13-month smoothed monthly total sunspot number (orange line in Fig. 5e, retrieved by the Royal Observatory of Belgium, http://www.sidc.be/silso/datafiles) is positively correlated with IMF7 ($r$ = 0.7). Signals of influence of the sunspot cycle in the upper troposphere-lower stratosphere have been indicated in several

works (e.g., Labitzke and Van Loon, 1997a, b; Coughlin and Tung, 2004), suggesting that the association between the Sun and stratospheric parameters (e.g., $O_3$) is due to solar-induced changes in the atmospheric circulation.

For Mt. Cimone the situation was different and more difficult to understand. The seasonal cycle is still present (IMF4, Fig. 6b), but IMF5 (Fig. 6c) exhibits no clear periodicity, presenting peaks at 20, 28 and 35 months. In this case, the comparison with the QBO signal does not highlight any evident relation. Variations in the peak amplitudes of IMF6 (Fig. 6d) are more regular, with two prominent oscillations with periods of 4.4 (ENSO) and 2.9 years. The last IMF (Fig. 6e) results in a characteristic period of 11.6 years, which most likely is related to the solar cycle, as indicated by the high correlation ($r = 0.8$) with the sunspot number (orange line in Fig. 6e).

## 5 Conclusions

In this work we presented a novel methodology (STEFLUX) to evaluate SI in a user-defined region, by using as input a Lagrangian STE climatology derived from the ERA-Interim reanalysis. Besides having shown an illustrative case study (Sect. 3.2) as a typical STEFLUX application, we investigated STEFLUX skills in detecting SI by comparing its time series with corresponding long-term SI time series derived from observational datasets (SIO, see Sect. 2). The analysis was performed in two very different areas of the world, i.e., the southern Himalayas (NCO-P) and the central Mediterranean basin (Mt. Cimone), which represent two global hot-spots for what concerns air pollution and climate change, and are often affected by SI.

Our results showed that STEFLUX correctly represented the typical seasonal cycles of SI frequencies over these two areas, with the highest occurrence of SI in winter for NCO-P, and in winter–spring for Mt. Cimone. For both sites, the lowest SI occurrence was recorded during summer (i.e., the monsoon for NCO-P). STEFLUX had real skill (higher for Mt. Cimone than NCO-P) in detecting single events at both regions, especially for robust (i.e., longer than 1 day) events. The identification of short events was more problematic; this is in agreement with a similar study by Cui et al. (2009), who reported considerable difficulties for two Lagrangian models in capturing "inconspicuous" SI events. This issue is also reflected by a low agreement in the evaluation of the inter-annual variability of SI frequency (especially for NCO-P during winter). Hence, STEFLUX should not be regarded as a tool to exactly reproduce SI occurrences at specific measurement sites, which typically are strongly affected by rather small-scale circulations. Instead, its premium application is in determining the SI input at a larger, regional scale. Moreover, although not investigated in this work, STEFLUX might be deployed as a particularly relevant tool to investigate how SI long-term variability influences the atmospheric composition at these specific locations (e.g., by deploying the $O_3$ values that are available along each trajectory).

The observed mismatch between STEFLUX and SIO is due to several reasons, such as the absence of representation of subgrid-scale processes in STEFLUX (e.g., convection, turbulent diffusion and mixing), along the path from the stratosphere to the target region. Last but not least, one should consider that these subgrid-scale processes can also lead to "local" or transient SI events captured by a single measurement point, which cannot be considered representative/significant for a whole region. In addition to this, another reason for the mismatch might be mixing and dilution processes occurring within air-masses from the tropopause crossing to the target region, expected to be maximized for greater travel times. Lastly, as also demonstrated in

previous studies, the "overpasses" phenomenon might have a not negligible impact. All of these discrepancies point out that a combination of modeling outputs (e.g., STEFLUX) and in situ observations is still needed to completely study, characterize and evaluate the occurrence of SI.

Another important feature of STEFLUX is its capability of climatologically assessing the SI occurrence at the chosen site, since the ERA-Interim reanalysis extends back to 1979. In this study, it allowed us to obtain a 35-year time series of SI events at NCO-P and Mt. Cimone, which affected 13% and 9% of the period, respectively. The tropopause crossings during the whole 35-y period, provided by STEFLUX, were further analyzed. NCO-P showed two main cluster regions, i.e., Central Asia (maximized for events with short travel times between the tropopause and the target box, $\Delta t$) and northern Africa (which had its maximum for events with 48 h $\leq \Delta t <$ 72 h). On the other hand, no preferred locations characterized Mt. Cimone, except for a small cluster south of Greenland during winter. We then evaluated trends on these long time series: no trends in the SI occurrence were discernible for both measurement sites, and the only statistically significant trend was observed for winter at NCO-P (0.18 % yr$^{-1}$). Furthermore, for the first time the CEEMDAN analysis has been performed on the time series characterizing these two hot-spot areas, in order to evaluate periodicities and their possible relation to climate factors. At NCO-P, signs of influence from the Quasi-Biennial Oscillation (QBO), the El-Niño/Southern Oscillation (ENSO) and the solar cycle were found, while Mt. Cimone exhibited relevant relations with ENSO and the solar cycle (high correlation with the sunspot number). These results indicate the possible impact of anthropogenic climate change on SI occurrence via changes in the ENSO and QBO regimes.

**STEFLUX availability**

The STEFLUX outputs are available on request by writing an e-mail to the authors, specifying the box characteristics and the period of study chosen.

*Acknowledgements.* The authors thank the ECMWF and MeteoSwiss for providing access to the meteorological data, the technical and logistic staff at NCO-P, F. Calzolari and F. Roccato (CNR–ISAC) for the technical support at Mt. Cimone, as well as the "Magera team" (C. Magera, P. Giambi and N. Gherardini) and the Italian Air Force (CAMM Monte Cimone) for the valuable co-operation. Furthermore, the authors thank the NOAA–CPC for the access to the MEI index, the Free University of Berlin for the QBO index, the Royal Observatory of Belgium for providing the sunspot number, and S. Pfahl for the useful discussions.

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

**Table 1.** Input parameters for STEFLUX for the comparison with in situ measurements (SIO).

| Parameter | NCO-P | Mt. Cimone |
|---|---|---|
| Lat_min, Lat_max | 26° N, 29° N | 43° N, 46° N |
| Lon_min, Lon_max | 85° E, 88° E | 9° E, 12° E |
| Box_top | 550 hPa | 790 hPa |
| Time span | 01 Mar. 2006–31 Dec. 2013 | 01 Jan. 1998–31 Dec. 2010 |
| Temporal resolution | 1 h | 1 h |

**Table 2.** 2×2 contingency tables for the comparisons of the SI events time series, i.e., identified by the SIO and STEFLUX approaches, for NCO-P (a, b) and Mt. Cimone (c, d). For each table the Odds Ratio Skill Score (ORSS) is also reported, along with the minimum ORSS required to have real skill at the 99% confidence level (in parentheses, see Thornes and Stephenson, 2001). Capital letters are defined as follows: A indicates the number of SI events selected by both methodologies (STEFLUX and SIO); B represents the number of events selected as SI by the first methodology but as no-SI by the second one; C represents the number of events selected as no-SI by the first methodology but as SI by the second one; and D represents the number of no-SI events selected by both methodologies.

| | | NCO-P | | | | | |
|---|---|---|---|---|---|---|---|
| (a) | | "SIO vs STEFLUX" | | (b) | | "STEFLUX vs SIO" | |
| | | STEFLUX | | | | SIO | |
| | | SI | no-SI | | | SI | no-SI |
| SIO | SI | A = 55 | B = 148 | STEFLUX | SI | A = 39 | B = 116 |
| | no-SI | C = 23 | D = 181 | | no-SI | C = 16 | D = 140 |
| ORSS | 0.49 (0.35) | | | ORSS | 0.49 (0.35) | | |
| | | Mt. Cimone | | | | | |
| (c) | | "SIO vs STEFLUX" | | (d) | | "STEFLUX vs SIO" | |
| | | STEFLUX | | | | SIO | |
| | | SI | no-SI | | | SI | no-SI |
| SIO | SI | A = 73 | B = 226 | STEFLUX | SI | A = 52 | B = 185 |
| | no-SI | C = 25 | D = 275 | | no-SI | C = 13 | D = 225 |
| ORSS | 0.56 (0.35) | | | ORSS | 0.66 (0.35) | | |

**Table 3.** (a) "SIO vs STEFLUX": agreement between STEFLUX and the measured SI events (SIO), and (b) "STEFLUX vs SIO": agreement between the measured and the modeled (by using STEFLUX) SI events, as a function of the different length of the SI events.

| (a) | NCO-P | | Mt. Cimone | |
|---|---|---|---|---|
| SI event duration | SI events by SIO | STEFLUX | SI events by SIO | STEFLUX |
| 1 day | 117 | 25 (22%) | 217 | 45 (21%) |
| 2 days | 41 | 17 (42%) | 36 | 11 (31%) |
| 3 days | 20 | 4 (20%) | 28 | 11 (39%) |
| ≥4 days | 25 | 9 (36%) | 18 | 6 (33%) |
| Total | 203 | 55 (27%) | 299 | 73 (24%) |

| (b) | NCO-P | | Mt. Cimone | |
|---|---|---|---|---|
| SI event duration | SI events by STEFLUX | SIO | SI events by STEFLUX | SIO |
| 1 day | 55 | 7 (13%) | 100 | 15 (15%) |
| 2 days | 36 | 15 (42%) | 73 | 23 (31%) |
| 3 days | 22 | 7 (32%) | 28 | 5 (18%) |
| ≥4 days | 42 | 10 (24%) | 36 | 9 (25%) |
| Total | 155 | 39 (25%) | 237 | 52 (22%) |

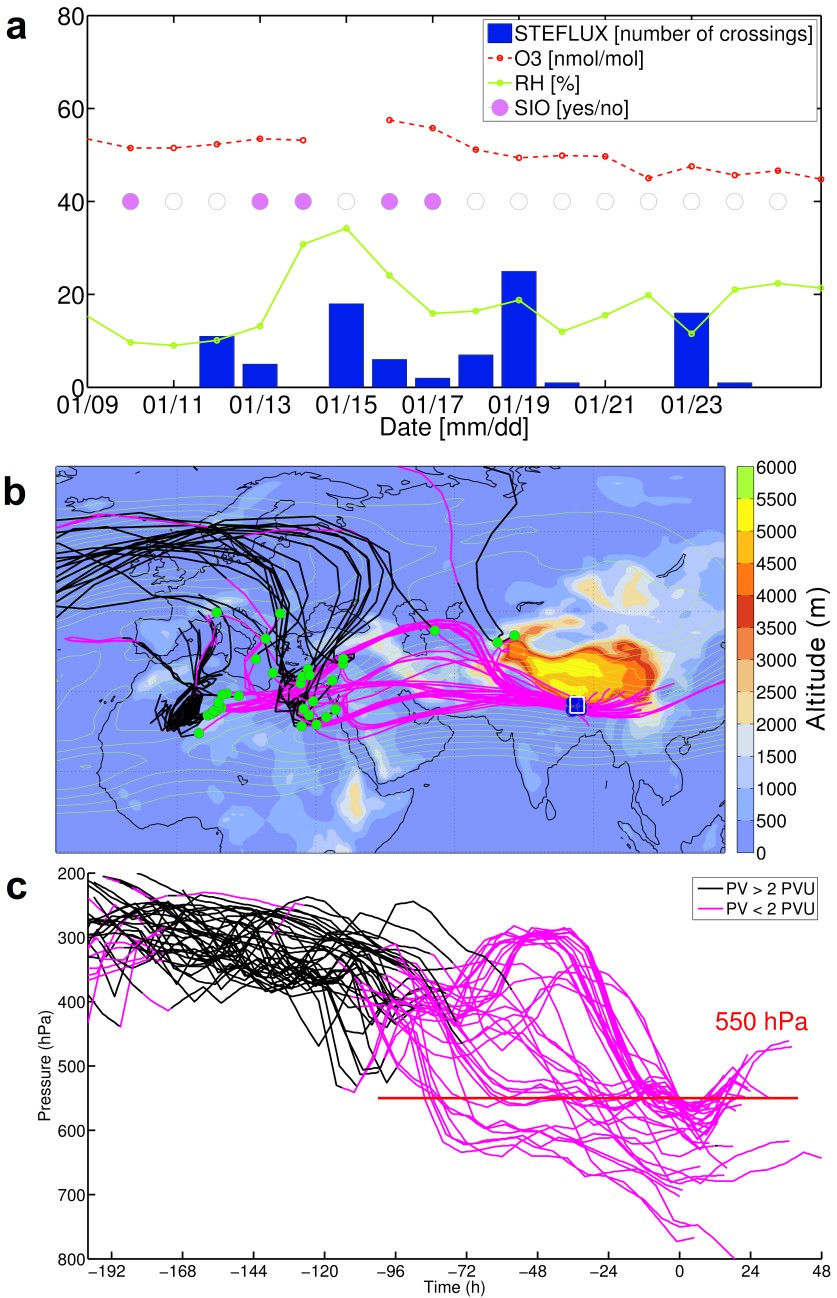

**Figure 1.** Example of application of STEFLUX, in a target box around NCO-P position over the period 09/01/2007–25/01/2007. Panel a shows the daily averages for O$_3$ and RH at NCO-P, the days selected by the SIO methodology (see Sect. 2) and the number of STE trajectory points crossing the box. The STE trajectories are also displayed entirely in panel b, where blue dots indicate the first crossings of the target box and green dots identify the tropopause crossing locations. Contour lines indicate the wind speed at 250 hPa, averaged over the case study period. Panel c shows the temporal-height evolution of the selected trajectories (0 is the time of arrival into the target box), as a function of the PV value along each trajectory point.

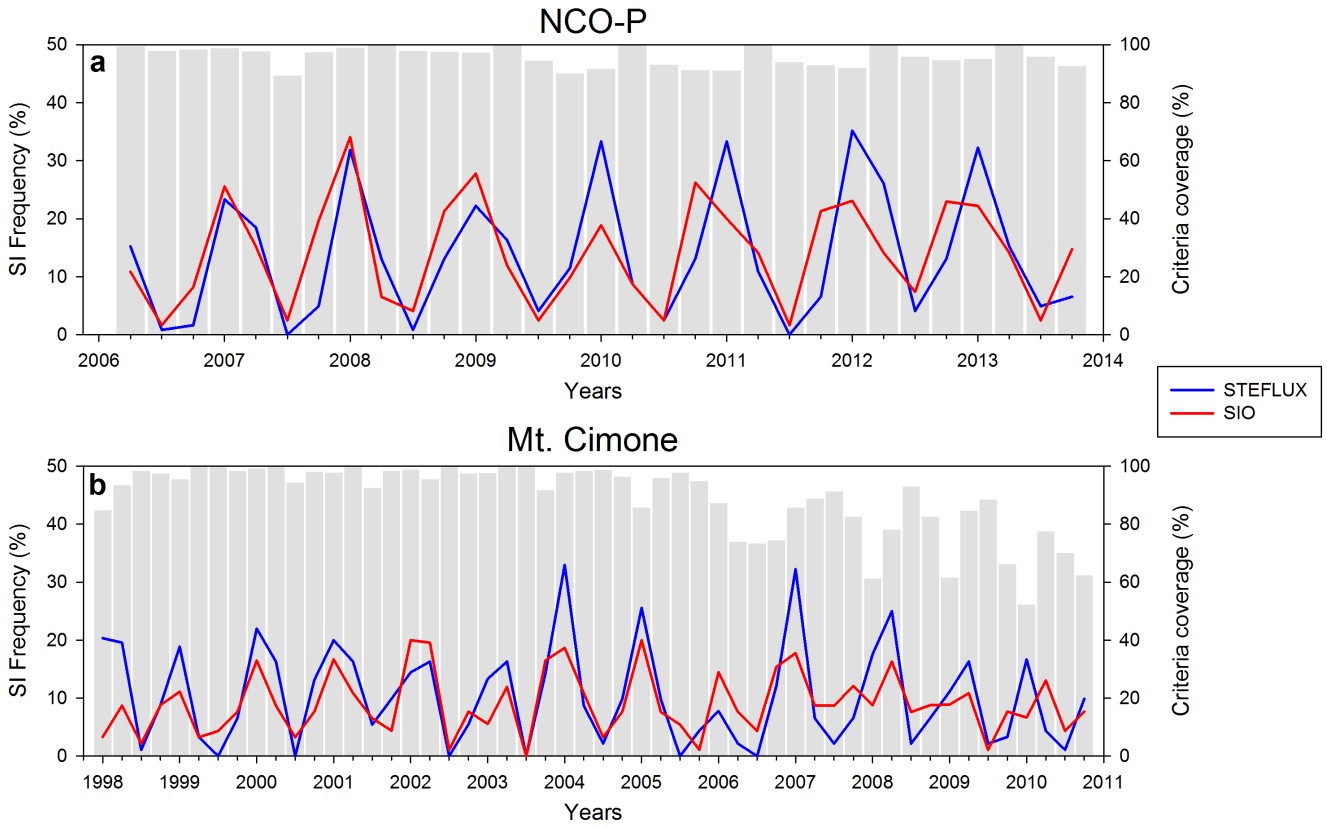

**Figure 2.** Seasonal graph of SI frequency at NCO-P (panel a) and Mt. Cimone (panel b). Blue lines indicate the STEFLUX outputs (for the configuration parameters see the Supplementary Material), while the red ones represent the results from the application of the criteria presented in Sect. 2 (SIO). Also shown in the plot is the percentage of criteria coverage (grey bars) for each season.

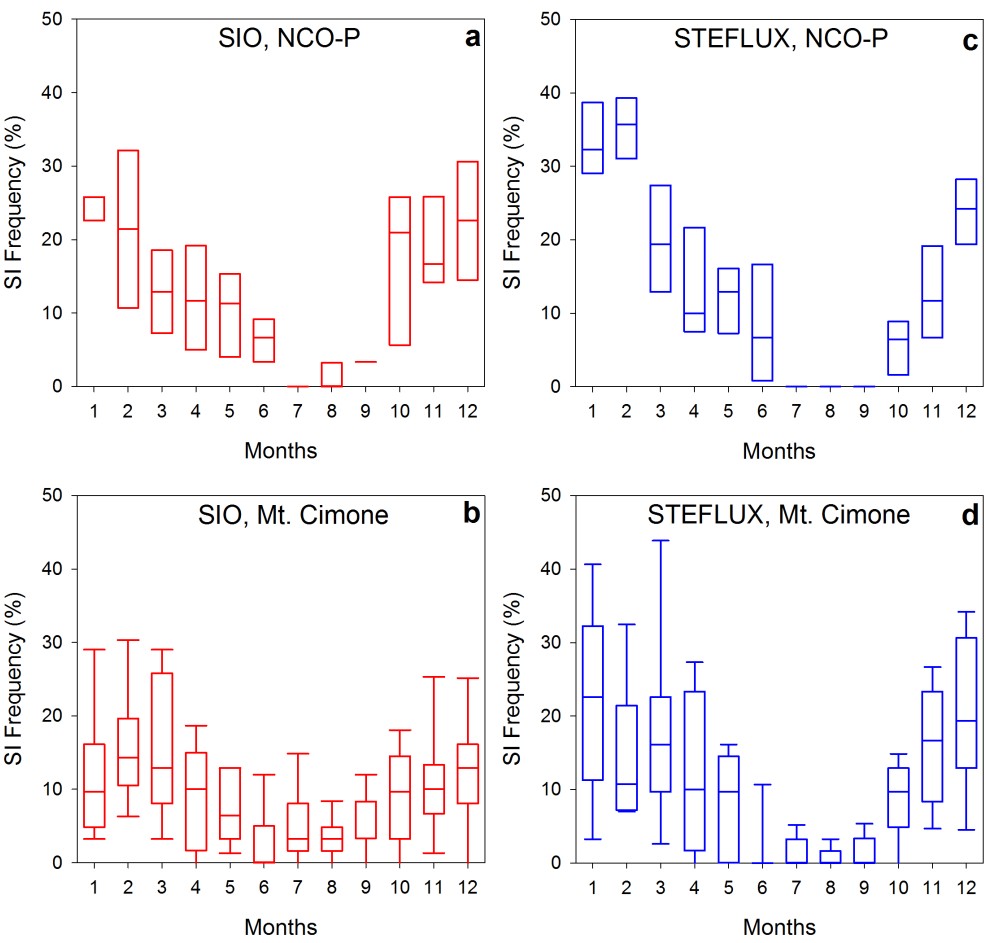

**Figure 3.** Box-whiskers plot of the annual variation of SI frequency at NCO-P (panels a,c) and Mt. Cimone (panels b,d). For both sites, the SIO (panels a,b) and STEFLUX (panels c,d) values are presented.

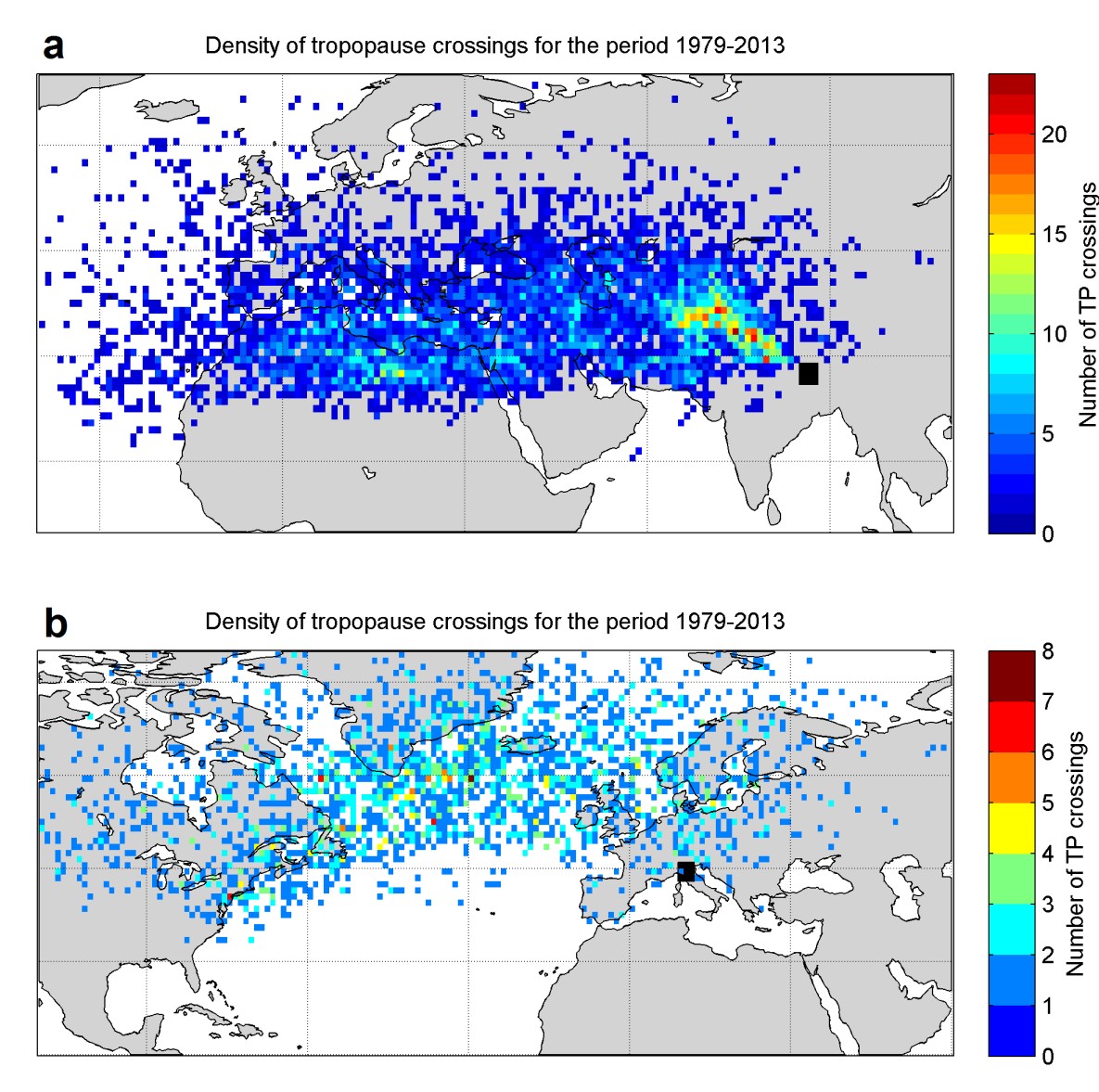

**Figure 4.** Density of tropopause (TP) crossings for the period 1979–2013, for NCO-P (panel a) and Mt. Cimone (b). Values for both figures were aggregated on a $1^\circ \times 1^\circ$ horizontal grid. In both panels, the black square indicates the horizontal extension of the STEFLUX box.

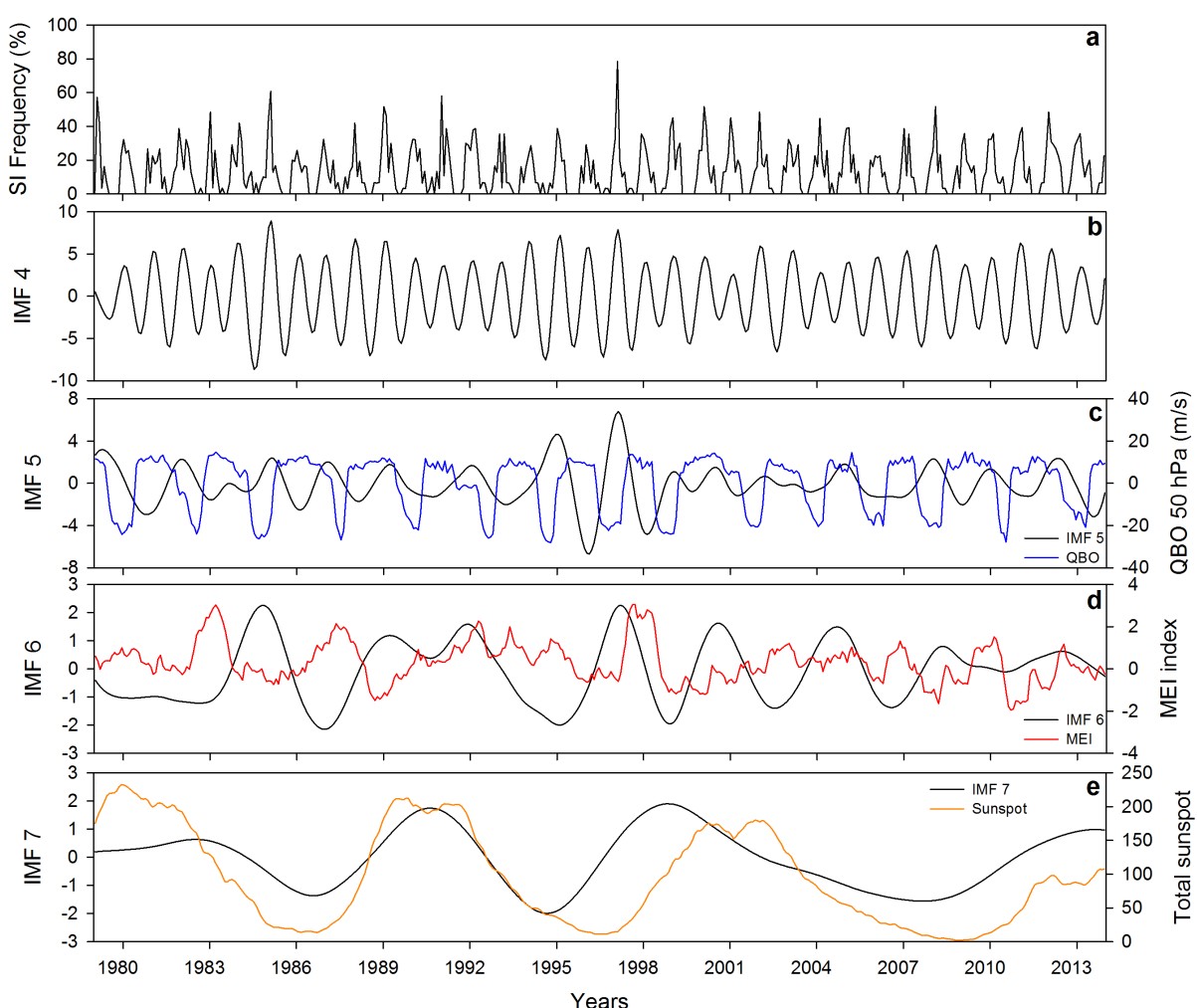

**Figure 5.** Time series of monthly averaged SI frequency, as retrieved by STEFLUX, at NCO-P (panel a), and some of its IMFs (i.e., IMF4–7, panels b–e, respectively) resulting from the application of the CEEMDAN analysis. The blue line in panel c represents the equatorial zonal wind at 50 hPa, used as a measure of the QBO signal, the red line in panel d depicts the Multivariate ENSO Index (MEI) and the orange line in panel e indicates the 13-month smoothed monthly total sunspot number.

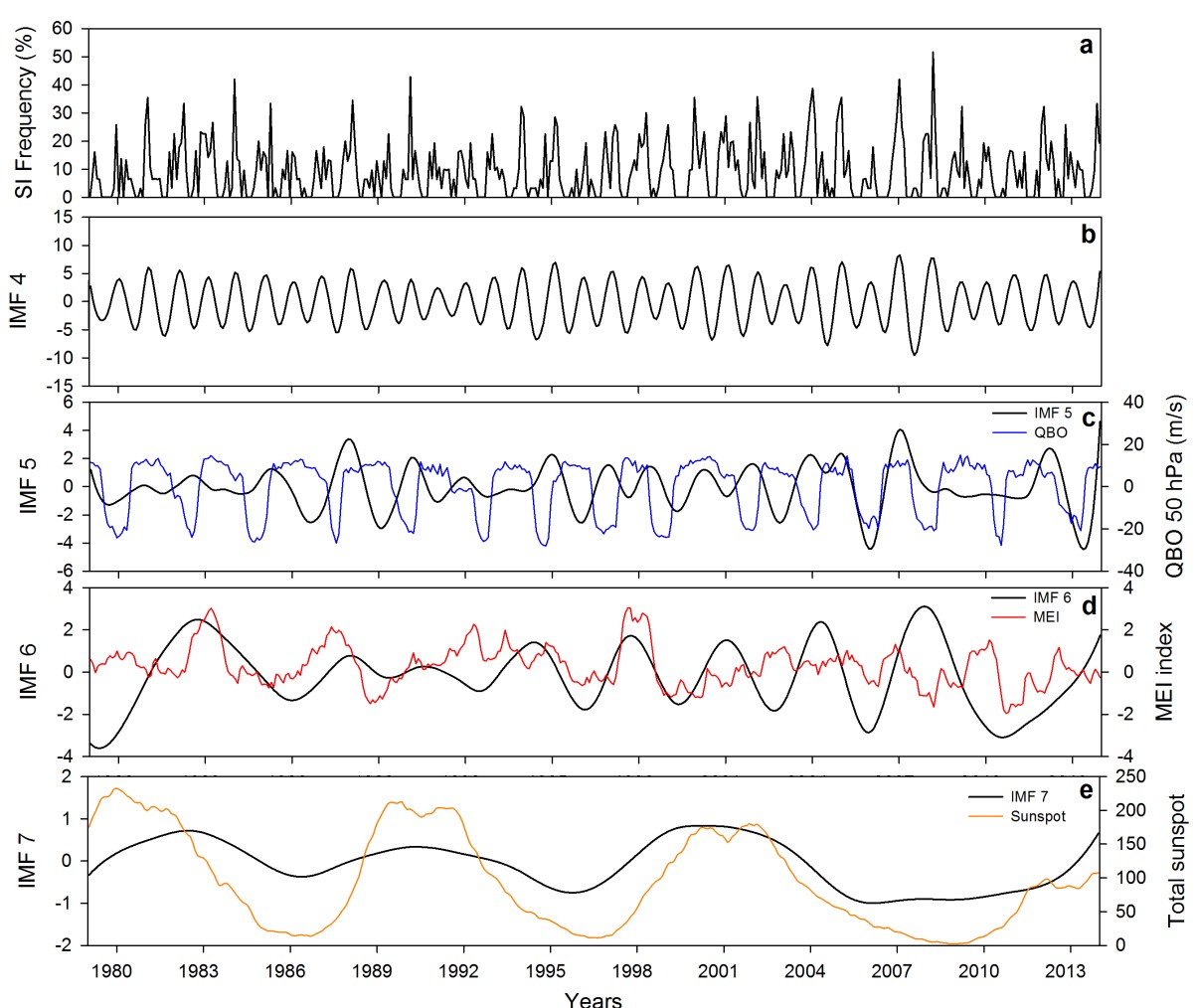

**Figure 6.** Same as Fig. 5, for Mt. Cimone time series.