# Peer review of "STEFLUX, a tool for investigating stratospheric intrusions: application to two WMO/GAW global stations"

_Atmospheric Chemistry and Physics, 2016_

## Referee Comment (RC1) · Anonymous Referee #1 · 24 Aug 2016

General Comments: This paper presents STEFLUX, a new tool that detects stratospheric intrusions affecting a specific location during a specific time period. STEFLUX is well described and the results are thoroughly discussed revealing the benefits and restrictions of the tool. As the transport of stratospheric air masses into the troposphere is of great importance, STEFLUX can be used in conjunction with observations for several scientific purposes. Therefore, I consider the paper to be an interesting study and recommend its publication in ACP, but only after addressing the following comments.

Main comments: It seems that there are inconsistencies between the skill scores values (False Alarm Rate, ORSS) presented in the manuscript (Page 7 line 27 – Page 8 line 5, Table 1) and the contingency tables presented in Table 1. Moreover, the presentation of the results in Table 1 needs to be more reader-friendly. I suggest the following:

1. Include (in Section 4.1.2) the formulas used to calculate all skill scores along with the corresponding references, i.e. ORSS=(AD-BC)/(AD+BC) (Thornes and Stephenson, 2001), explaining what A, B, C and D stands for in your case.

2. Assign A, B, C and D to the respective values in Table 1.

3. Check calculations for the skill scores. It is likely that your results are better (higher ORSS values and lower False Alarm Rate values).

4. Add a label for each table in Table 1, in order to be clear which approach is the "reference" and which is the "predictor". i.e for Table 1a,c "SIO vs STEFLUX" and for Table 1b,d "STEFLUX vs SIO".

5. Revise the discussion (for skill scores) in the manuscript if needed.

Minor comments: Please add degree symbols for lon and lat values in the manuscript. i.e. Page 3, line 20.

Section 4.1.1: Please include a definition for STEFLUX SI day. i.e. threshold of at least 1 box crossing per day?

Page 7, lines 6-7: "(see the Supplementary Material)". Please specify exactly where in the Supplementary Material.

Table 2: Add "(b)" in the second table.

Figure 1a: Please replace "STEFLUX [#]" with "STEFLUX [number of crossings]".

Figure 4: The map continents are not so clear. Please change map continents color if possible (maybe grey).

Thornes, J. E. and Stephenson, D. B.: How to judge the quality and value of weather forecast products, Meteorological Applications, 8, 307–314, 2001.

---

## Referee Comment (RC2) · Anonymous Referee #2 · 26 Aug 2016

In this study, the authors present a new tool, called STEFLUX, to select trajectories having crossed the tropopause downward at some time in the past days and arriving into a user-defined geographical box within a prescribed time-window. The trajectories are selected among a large set of pre-computed trajectories based on the ERA-Interim reanalysis from the ECMWF. Doing this, this is presumably a fast-computing tool since no trajectory computation is needed.

Output data allow for various applications, such as assessing the occurrence frequency of stratospheric intrusions (SI) in the lower troposphere at any place on Earth at regional scale, but also characterizing preferred entry regions in the UTLS, travel times until the target area, etc. The paper presents an illustrative case study, a skill assess-

ment study with respect to SI detection based on (mainly ground-based) observations, and finally a climatology over 35 years of SI events over two focal areas.

STEFLUX is certainly a promising tool which may be helpful for a scientific community larger than the authors' research team. The paper itself is fairly well-built and written, and the presented scientific material and discussions are of good scientific quality.

Therefore, I recommend the publication of this study in ACP, but not before the author take in consideration the following comments and propose a revised version of their manuscript. I would appreciate if the authors could pay particular attention to my general comments 3 and 4.

**General comments**

*1. Method originality not fully clear*

While reading Section 1, it is not straightforward to know what is new in the STEFLUX method compared to existing methods based on backtrajectories. For instance, one could wonder why don't the author simply initialize backtrajectories from the target regions and see if their cross the tropopause at some time in the past?

I guess one major advantage of the method is computation speed, and this is due to the fact that it works from pre-computed backtrajectories. But this is not clearly stated in the text.

More generally, I think the Introduction could be developed and depict more explicitly the state-of-the-art in the domain: what are the different types approaches? what are their drawbacks or limitations? etc. The originality of the STEFLUX method should thus be more emphasized.

*2. Representativity of a deep valley station*

At several places in the text, it is suggested (e.g. when mentioning the "overpass" effect) that SIO may be missed at the surface stations because their measurement may not always be representative of the free troposphere at regional scale owing to local mountain meteorology. I think this concern is especially true for the NCO-P station, which is located in the bottom of a deep valley. Even in conditions of down-valley flow, it is likely that air has been in contact with the surface before reaching the observatory. Ozone in particular may have experienced deposition, and surface ozone concentrations may be lower than those encountered in the free troposphere. Valleys are indeed known to be net sinks for ozone (see e.g. Furger et al., Atmos. Env., 34, 1395-1412; Wotawa and Kromp-Kolb, Atmos. Env., 34, 1319-1322).

Even in the cited references (Bonasoni et al., 2010; Cristofanelli et al. 2010) little is said on the station representativeness at regional scale (except in the monsoon season at night). It would be worthy if this question could be briefly discussed somewhere in the paper (e.g. in Section 2 when the station are presented).

In contrast, I am much more confident in the regional representativeness of the mountain-top site Monte-Cimone (of course, apart from anabatic conditions) and the suitability of the site to detect deep stratospheric intrusion, although it is at much lower altitude.

*3. SIO detection criteria too imprecise*

In Section S1.4 (supplementary material), the SIO selection criteria are presented in a too vague and qualitative manner (and therefore the criteria appear to be subjective). For instance, what does "significant variation of daily P" mean? What is the threshold to consider the variation is significant? Further, is the current pressure daily mean compared to the value the day before?

One could ask such questions for almost every items of the two lists. The authors must present their study in a reproducible way, and those criteria are central elements. This

section should be rewritten in a much more rigourous and quantitative manner, with the concern of study reproducibility.

*4. Missing discussion on backtrajectory maximum duration*

In this study, tropopause crossings are considered up to 5 (= 1+4) days prior the trajectories reach the target box. But if one goes sufficiently deep backward in time, any trajectory ending in the target box crossed the tropopause at some time in the past. On the contrary, if the trajectory maximum duration is reduced below some value, no SI at all is detected.

Actually, the target region can be found to be from 0% to 100% of the time under the influence of stratospheric intrusions, depending on the chosen trajectory maximum duration. This parameter appears to be of central importance in the STEFLUX tool. I think a sensitivity study to this parameter should be presented (especially in relation with the results (percentages) given in Section 4.2.1), or at least, the choice of 5 days (which obviously comes from the work by Skerlak et al., 2014) should be carefully discussed and justified.

This leads to a more general question: any sufficiently long-lived molecule in the troposphere resided in the stratosphere at some prior times. What is the typical lifetime of a stratospheric intrusion in the troposphere, and when should one consider the air mass composition as being no longer influenced by the stratosphere?

I think these points are crucial in this study and deserve thorough discussions.

*5. Links with ENSO, QBO and sunspots poorly convincing*

In Section 4.2.2, the authors claim that some IMFs are correlated with various indicators (of ENSO, QBO, solar activity), but I find that Figure 5 and 6 poorly support these results (at least when examined by eye). Could these correlations be demonstrated more clearly, for instance by means of scatterplots?

Beyond this, correlation is not causality. A correlation is interesting to consider only if one suspects some mechanism linking two quantities. In the text, the possible link between ENSO and STE is discussed, but to a much lesser extent the links with the QBO and the solar activity. Could the authors discuss or even speculate a bit more about this?

*6. Balance between paper main body and supplementary material*

The article main body is quite concise in it present form, and I think there is perhaps room for moving important elements from the supplementary material into the paper main body.

For instance, the criteria to detect SIO are of primary importance in the study and could appear in the article, as well as Table S1, and perhaps also Figures S4 and S5.

**Specific comments**

p.1, l.2: The use of upper-case letters suggests that "STEFLUX" is an acronym. In this case, could the authors make it explicit at least once in the abstract and in the main text body? If it is no acronym but a simple proper noun, I suggest one should write "Steflux".

p.1, l.19: Please consider to change "relating" by "linking".

p.1, l.9-10: "show still" → "still show".

p.2, l.14, "anticyclonic": Do the author mean "cyclonic" instead?

p.2, l.17, "due to anthropogenic emissions": I would specify: local or regional. Please also consider that local or regional biogenic emissions may also alter atmospheric composition with respect to the tropospheric background.

p.2, l.17: "make" → "makes".

p.2, l.27, "Many different methods are based on this combined approach (...) and vary considerably between different measurement sites.": These statements are supported by no literature reference. Could the author cite here a list of references or at least a review paper on the topic? What are those considerable variations between the method? Could the author be a bit more explicit? See also my general comment 1.

p.2, l.27, "occurring over": reaching? detected?

p.2, l.33-34, "Moreover, ...": It seems that this potential application is not illustrated in the paper. Could the author justify this statement?

p.3, l.5: "to it" → "on climate".

p.3, l.19-20: This statement is questionable and deserves further discussion. See my general comment 2.

p.3, l.24-25, "starting at the measurement site": this is too imprecise, especially concerning the altitude. Was the true site altitude or the model surface altitude used to initialize the backtrajectories?

p.4, whole Section 3.1: even though the case study clarifies well what STEFLUX is (Sect. 3.2), Section 3.1 presenting the tool is confusing. Especially, it is hard to distinguish what comes from Skerlak et al. and what is specific to STEFLUX. Beyond this, a number of elements from Skerlak et al.'s methodology are mentioned in the text (trajectories extended 4 days prior to tropopause crossing; 3D labeling) but it seems these details are not needed in STEFLUX or at least in this paper. If really not needed, these information items are confusing and should be removed. Otherwise, it should be explained why they are important. More generally, I think that the whole Section 3.1 should be rewritten and clarified.

p.4: title of Section 3.2 could be changed to "Illustrative case study".

p.4, l.29: The box centered at NCO-P is hardly visible in Fig.1b. Anyway, a reference to this Figure is not useful in this sentence, and mention to Fig.1b could be simply

removed here.

p.4, l.30: "recorded" can be removed.

p.5, l.5: in the present form of the paper, the criteria are actually introduced in the supplementary material, not in Section 2. See also my general comment 6.

p.5, l.11 and ff.: it seems from these lines that there are three different output files from a STEFLUX run, but it is not clear what is in those files. This should be clarified (perhaps in Section 3.1).

p.5, l.17: "indicated in previous studies ..." → "identified as a preferred region for tropopause crossing in previous studies ...".

p.5, l.26, "they still maintained a stratospheric signature": poor expression, please rephrase.

p.5, l.33: the choice of an horizontal extension of $3° \times 3°$ should be justified briefly.

p.6, l.2-3, "The selected time periods were the same as in Sect.2": please specify.

p.6, l.4: "a table listing ..." → "Table S1 listing ...".

p.6, Section 4.1.1: What is the criterion to tag a day as SI day according to STEFLUX? Is only one box crossing at any moment of the day and of any duration needed? The author should specify this in this Section. (See also the corresponding comment from the Anonymous Referee #1.)

p.6, l.11, "at the two measurement sites": not needed and a bit confusing, please remove.

p.6, l.24: "subtle" is unexpected as adjective for the inter-annual variability. Please rephrase.

p.7, l.1, "criteria coverage": please define. Is it the fraction of time when the data used in the criteria are simultaneously available? Every criterion does not use all the data:

what does happen when one data is missing for one criterion but another criterion is fulfilled? Or none other fulfilled? Is the day tagged as SI/non-SI day or discarded? Please clarify.

p.7, l.22 and ff.: I had a hard time to understand those contingency tables. Considering for instance Table 1(a), does 55 means that during 55 SIO events, STEFLUX detected more than 50% of time of the episode as SI? Does 148 means that during 148 SIO events, STEFLUX detected less than 50% of time of the episode as SI? etc. Please explain a bit more how those numbers should be interpreted. See also the comment from the Anonymous Referee #1 concerning the definitions of accuracy and false alarm rate: how exactly are the presented scores calculated?

p.8, l.2 and 5: the capture rates given in Table 2 (22-27%) are closer to one quarter than to one third.

p.8, l.20-24: In case of long travel time and high mixing, can one still consider the air mass as a stratospheric intrusion? See my general comment 4 on stratospheric intrusion lifetime.

p.9, l.10: this again is related to my general comment 4: is it really relevant to be irrespective of the degree of mixing and dilution in the troposphere?

p.9, l.23-25: could the author explain this statement?

p.9, l.32: "If divided seasonally" → "Considering seasons separately"

p.9, l.33 and p.12 l.15: "southward of" → "south of"

p.11, l.12 "does not exhibit as" → "exhibits no"

p.11, l.18 "defined" → "user-defined"

p.11, l.19 "representative" → "illustrative"

p.12, l.17 "both of the" → "both"; "significant" → "statistically significant"

p.13, Appenzeller and Davies, 1992: insufficient reference.

p.16, Table 2: missing "(b)".

p.18, figure legend, l.3: "Sect. 2" → "Sect. S1.4". See my general comment 6.

p.19, figure 3: the STEFLUX and SIO panel columns could be interchanged, so that the panels (a-d) are numbered in the same order as in the text. Why do the box plots in the upper panel have no whiskers?

*Supplementary material*

p.1, l.18: do the authors mean gamma-spectroscopy?

p.1, l.25: "total column OF ozone".

p.2, l.9-10: This sentence is not fully clear. What does "centered at an ending altitude" mean? Is 490hPa the real altitude of NCO-P? In the same vein in l.12, do the trajectories reach Mt. Cimone at its real altitude level? What is the corresponding pressure level?

p.2, Section S1.4 (SI selection criteria): see my general comments 3 and 6.

---

## Author Comment (AC1) · 6 Oct 2016

General Comments: This paper presents STEFLUX, a new tool that detects stratospheric intrusions affecting a specific location during a specific time period. STEFLUX is well described and the results are thoroughly discussed revealing the benefits and restrictions of the tool. As the transport of stratospheric air masses into the troposphere is of great importance, STEFLUX can be used in conjunction with observations for several scientific purposes. Therefore, I consider the paper to be an interesting study and recommend its publication in ACP, but only after addressing the following comments.

*We thank the Reviewer for his/her valuable suggestions and his/her encouraging evaluation. In the following, we report our point-to-point replies to each of the raised points.*

[Figure]

*Modifications to the text are performed in the revised version of the manuscript and are marked in red and blue colors in the track-changes manuscript version.*

Main comments: It seems that there are inconsistencies between the skill scores values (False Alarm Rate, ORSS) presented in the manuscript (Page 7 line 27 – Page 8 line 5, Table 1) and the contingency tables presented in Table 1. Moreover, the presentation of the results in Table 1 needs to be more reader-friendly. I suggest the following:
1. Include (in Section 4.1.2) the formulas used to calculate all skill scores along with the corresponding references, i.e. ORSS=(AD-BC)/(AD+BC) (Thornes and Stephenson, 2001), explaining what A, B, C and D stands for in your case.
2. Assign A, B, C and D to the respective values in Table 1.
3. Check calculations for the skill scores. It is likely that your results are better (higher ORSS values and lower False Alarm Rate values).
4. Add a label for each table in Table 1, in order to be clear which approach is the "reference" and which is the "predictor". i.e for Table 1a,c "SIO vs STEFLUX" and for Table 1b,d "STEFLUX vs SIO".
5. Revise the discussion (for skill scores) in the manuscript if needed.

*We thank the Reviewer for this important comment, and we apologize for the inconsistencies between the wrong values given in the text and those reported in Table 1. Our new results are indeed better than those presented in the first version of the manuscript, with a lower false alarm rate (equal to 0.45), and higher ORSS (see the updated Table 1) for all comparisons. These values have been updated in the text (Page 8, Lines 24–25) and in Table 1.*
*Moreover, as suggested by the Reviewer, we made the description of the skill scores and the layout of Table 1 clearer. Formulas for the accuracy, false alarm rate and ORSS have been inserted (following Thornes and Stephenson, 2001), explaining also what A, B, C and D stand for (see Sect. 4.1.2). Table 1 has been updated by specifying what these capital letters refer to and by adding labels for the "STEFLUX vs SIO" or "SIO vs STEFLUX" comparisons. The caption has also been modified accordingly.*

Minor comments: Please add degree symbols for lon and lat values in the manuscript. i.e. Page 3, line 20.

*Done.*

Section 4.1.1: Please include a definition for STEFLUX SI day. i.e. threshold of at least 1 box crossing per day?

*The definition of SI day was erroneously given in Sect. 4.1.2 (Page 7, Lines 10–11): "…SI days (with a threshold of at least 2 box crossings per day for STEFLUX, in order to retain robust information only and to discharge "erratic" events).". Thus, this sentence has been moved above (Page 7, Lines 17–18).*

Page 7, lines 6-7: "(see the Supplementary Material)". Please specify exactly where in the Supplementary Material.

*Done. Since old Table S1 has been moved into the main body of the text (see our answer to Reviewer #2 comments), the sentence has been modified to: "(see Table S1 in the Supplementary Material)".*

Table 2: Add "(b)" in the second table.

*Done.*

Figure 1a: Please replace "STEFLUX [#]" with "STEFLUX [number of crossings]".

*Done.*

Figure 4: The map continents are not so clear. Please change map continents color if possible (maybe grey).

*Done.*

---

## Author Comment (AC2) · 6 Oct 2016

In this study, the authors present a new tool, called STEFLUX, to select trajectories having crossed the tropopause downward at some time in the past days and arriving into a user-defined geographical box within a prescribed time-window. The trajectories are selected among a large set of pre-computed trajectories based on the ERA-Interim reanalysis from the ECMWF. Doing this, this is presumably a fast-computing tool since no trajectory computation is needed.

Output data allow for various applications, such as assessing the occurrence frequency of stratospheric intrusions (SI) in the lower troposphere at any place on Earth at regional scale, but also characterizing preferred entry regions in the UTLS, travel times

until the target area, etc. The paper presents an illustrative case study, a skill assess-ment study with respect to SI detection based on (mainly ground-based) observations, and finally a climatology over 35 years of SI events over two focal areas.

STEFLUX is certainly a promising tool which may be helpful for a scientific community larger than the authors' research team. The paper itself is fairly well-built and written, and the presented scientific material and discussions are of good scientific quality.

Therefore, I recommend the publication of this study in ACP, but not before the au-thor take in consideration the following comments and propose a revised version of their manuscript. I would appreciate if the authors could pay particular attention to my general comments 3 and 4.

*We thank the Reviewer for his/her valuable suggestions and his/her encouraging eval-uation. In the following, we report our point-to-point replies to each of the raised points. Modifications to the text are performed in the revised version of the manuscript and are marked in red and blue colors in the track-changes manuscript version.*

**General comments**

*1. Method originality not fully clear*

While reading Section 1, it is not straightforward to know what is new in the STEFLUX method compared to existing methods based on backtrajectories. For instance, one could wonder why don't the author simply initialize backtrajectories from the target regions and see if their cross the tropopause at some time in the past?

I guess one major advantage of the method is computation speed, and this is due to the fact that it works from pre-computed backtrajectories. But this is not clearly stated in the text. More generally, I think the Introduction could be developed and depict more explicitly the state-of-the-art in the domain: what are the different types approaches? what are their drawbacks or limitations? etc. The originality of the STEFLUX method should thus be more emphasized.

*As correctly pointed out by the Reviewer, one characteristic of STEFLUX is the com-*

*putational speed, since no calculation of back-trajectories is required. This would be a particularly long and time-consuming task, especially when evaluating STE over long periods. The calculation of backward trajectories from the target point would have been possible for NCO-P and Mt. Cimone, but the aim of STEFLUX is to be a tool that can be easily applied at any point on the globe. Therefore, it is valuable to have a set of pre-computed forward trajectories (please refer to Škerlak et al., 2014, to see how the trajectories are generated), because a simple calculation of backward trajectories is not feasible. Also, as demonstrated in the paper, it is sufficient specifying only a few parameters (spatial coordinates and top lid of the box) in STEFLUX, for obtaining a quick and reliable estimate of the STE occurred over the desired time window. In addition to this, being based on the constantly updated ERA-Interim reanalysis, STEFLUX allows for a STE estimate back to 1979, thus it is especially important from a climatological perspective. In order to highlight these motivations in the text, we modified as follows (Page 3, Lines 7–8): "The tool, called STEFLUX (Stratosphere-to-Troposphere Exchange Flux), is a relatively fast-computing algorithm which makes use of the pre-computed trajectories composing the STE climatology by. . ." and (Page 3, Lines 12–14): "Its computational speed and user-friendly approach (it is sufficient to specify only a few parameters to work) make it suitable for obtaining a quick and reliable estimate of the SI occurred at a specific place over the desired time window (including long periods which would otherwise require a lot of time-consuming calculations). A potential. . .". Moreover, the Introduction part concerning the different "observations-based" methodologies to detect STE has been developed including several works, together with one review paper (Pages 2–3): "Usually, stratospheric influence is detected at a measurement site by analyzing the variability of in situ "stratospheric" observations (e.g., relative humidity, $^7Be$, $^{10}Be$, $O_3$, atmospheric pressure variability) and profiling datasets (radio/ozone-sondes), coupled with the analysis of satellite (e.g., total column of ozone) and various kinds of numerical weather prediction (NWP) model products fields. Many different methods, as thoroughly reviewed in Stohl et al. (2003), are based on this combined approach. Stohl et al. (2000) deployed a detection algorithm based on*

*the in situ variation of experimental data and simulations with a passive stratospheric tracer. Similarly, other studies analyzed STE by coupling experimental data and back-trajectories (e.g., Cristofanelli et al., 2006, 2010; Trickl et al., 2010). Usually, specific threshold values are applied to in situ tracers' variability to detect the presence of air-masses with stratospheric "fingerprints". Also trajectory and dispersion models are extensively used to detect the occurrence of STE. For example, Cui et al. (2009) used the particle dispersion model FLEXPART (Stohl et al., 2005) and the trajectory model LAGRANTO (Wernli and Davies, 1997) to identify stratospheric transport at the high-altitude Alpine site Jungfraujoch (Switzerland), while Tarasova et al. (2009) deployed 3D air-mass back-trajectories to trace the atmospheric transport at two high mountain measurement sites over the Alps and Caucasus. As pointed out by Bourqui (2006), trajectory-based approaches can provide a lower-bound estimate for STE flux, while dispersion models can provide slightly larger estimates. Typically, when used to detect STE at specific locations at the Earth's surface, all of these "observations-based" methodologies vary among different measurement sites, with respect to the number and types of stratospheric tracers available/considered, threshold values adopted, and often require a lot of time-consuming implementation to work. Moreover, it should be argued that none of the most diffused tracers have a "pure" stratospheric origin; for example, $^7Be$ and $O_3$ are affected by significant tropospheric sources. Furthermore, the compilation of proper long-term climatologies is very often hindered by the lack of long-term observations of "stratospheric" tracers.".*

*Stohl, A., Forster, C., Frank, A., Seibert, P., and Wotawa, G.: Technical note: The Lagrangian particle dispersion model FLEXPART version 6.2, Atmos. Chem. Phys., 5, 2461-2474, 2005.*
*Tarasova, O. A., Senik, I. A., Sosonkin, M. G., Cui, J., Staehelin, J., and Prévôt, A. S. H.: Surface ozone at the Caucasian site Kislovodsk high mountain station and the Swiss Alpine site Jungfraujoch: data analysis and trends (1990–2006). Atmos. Chem. Phys., 9, 4157-4175, 2009.*

[Figure]

**2. Representativity of a deep valley station**

At several places in the text, it is suggested (e.g. when mentioning the "overpass" effect) that SIO may be missed at the surface stations because their measurement may not always be representative of the free troposphere at regional scale owing to local mountain meteorology. I think this concern is especially true for the NCO-P station, which is located in the bottom of a deep valley. Even in conditions of down-valley flow, it is likely that air has been in contact with the surface before reaching the observatory. Ozone in particular may have experienced deposition, and surface ozone concentrations may be lower than those encountered in the free troposphere. Valleys are indeed known to be net sinks for ozone (see e.g. Furger et al., Atmos. Env., 34, 1395-1412; Wotawa and Kromp-Kolb, Atmos. Env., 34, 1319-1322).

Even in the cited references (Bonasoni et al., 2010; Cristofanelli et al. 2010) little is said on the station representativeness at regional scale (except in the monsoon season at night). It would be worthy if this question could be briefly discussed somewhere in the paper (e.g. in Section 2 when the station are presented).

In contrast, I am much more confident in the regional representativeness of the mountain-top site Monte-Cimone (of course, apart from anabatic conditions) and the suitability of the site to detect deep stratospheric intrusion, although it is at much lower altitude.

*The Reviewer raised an interesting point concerning the representativeness of NCO-P ozone observations. Unfortunately, no specific studies aimed at assessing these kinds of processes at NCO-P have been carried out so far. For this reason, in Sect. 4.1.3, the following sentences have been added, as well as two new references (Page 10, Lines 15–19): "Furthermore, for NCO-P, it should be considered that the station is located in a narrow valley. Thus, it is conceivable that, during the transport within the valley, $O_3$ (one of the stratospheric tracers considered by SIO) experiences deposition phenomena, thus decreasing the actual concentration that the stratospheric air-mass would have in the free troposphere (see, e.g., Furger et al., 2000; Wotawa and Kromp-Kolb, 2000).".*

*Furger, M., Dommen, J., Graber, W. K., Poggio, L., Prévôt, A. S., Emeis, S., Grell, G., Trickl, T., Gomiscek, B., Neininger, B., and Wotawa, G.: The VOTALP Mesolcina Valley Campaign 1996 – concept, background and some highlights, Atmos. Environ., 34, 1395-1412, 2000.*

*Wotawa, G., and Kromp-Kolb, H.: The research project VOTALP – general objectives and main results, Atmos. Environ., 34, 1319-1322, 2000.*

*3. SIO detection criteria too imprecise*

In Section S1.4 (supplementary material), the SIO selection criteria are presented in a too vague and qualitative manner (and therefore the criteria appear to be subjective). For instance, what does "significant variation of daily P" mean? What is the threshold to consider the variation is significant? Further, is the current pressure daily mean compared to the value the day before?

One could ask such questions for almost every items of the two lists. The authors must present their study in a reproducible way, and those criteria are central elements. This section should be rewritten in a much more rigourous and quantitative manner, with the concern of study reproducibility.

*We apologize for having provided too little detail when describing the SIO criteria. The significant variations of the several parameters (TCO, P, $^7$Be) are computed by the following methodology: first, a three-time repeated iteration of a 21-days running mean (the so-called Kolmogorov-Zurbenko filter, see Sebald et al., 2000) is applied to the daily average time series; then, residuals are computed by subtracting these latest values from the daily averages; residuals which exceed the upper (for $^7$Be) or upper and lower (for TCO and P) endpoints of the 95% confidence interval of the residuals distribution over the whole period are thus labelled as "significant variations".*

*In order to make the SIO selection methodology clearer and available in the main text, Sect. S1.4 has been rewritten including these detailed information and moved to Sect. 2.*

*Sebald, L., Treffeisen, R., Reimer, E., and Hies, T.: Spectral analysis of air pollutants.*

*Part 2: ozone time series, Atmos. Environ., 34, 3503-3509, 2000.*

*4. Missing discussion on backtrajectory maximum duration*
In this study, tropopause crossings are considered up to 5 (= 1+4) days prior the trajectories reach the target box. But if one goes sufficiently deep backward in time, any trajectory ending in the target box crossed the tropopause at some time in the past. On the contrary, if the trajectory maximum duration is reduced below some value, no SI at all is detected.

Actually, the target region can be found to be from 0% to 100% of the time under the influence of stratospheric intrusions, depending on the chosen trajectory maximum duration. This parameter appears to be of central importance in the STEFLUX tool. I think a sensitivity study to this parameter should be presented (especially in relation with the results (percentages) given in Section 4.2.1), or at least, the choice of 5 days (which obviously comes from the work by Skerlak et al., 2014) should be carefully discussed and justified.

This leads to a more general question: any sufficiently long-lived molecule in the troposphere resided in the stratosphere at some prior times. What is the typical lifetime of a stratospheric intrusion in the troposphere, and when should one consider the air mass composition as being no longer influenced by the stratosphere?

I think these points are crucial in this study and deserve thorough discussions.

*The time aspect is always a difficult task in trajectory analyses. The typical lifetime of a stratospheric intrusion in the troposphere has been considered in several papers (e.g., Stohl et al., 2000; Bourqui and Trépanier, 2010; Trickl et al., 2014, 2016). Stohl et al. (2000) argue that, once brought into the troposphere, the stratospheric signature of an air-mass (i.e., low RH, high $O_3$) gets lost over the period of a few days, because it gets quasi-adiabatically stirred by large-scale cyclonic and anticyclonic disturbances. The choice of 5 days for trajectories is also presented in Bourqui and Trépanier (2010). It was found that the trajectory clusters for their case studies experienced three distinct phases during their descent from the stratosphere (namely crossing of the tropopause,*

*free descent, and quasi-horizontal dispersion in the troposphere), and this whole pro-
cess takes typically 4-5 days. Similarly, Trickl et al. (2016) define "stratospheric intru-
sion trajectories" as those initially residing in the stratosphere and descending during
the following 5 days by more than 300 hPa into the troposphere. Finally, we could jus-
tify our choice with a simple order-of-magnitude calculation. If we suppose that an air
parcel descends uniformly from the tropopause (∼10 km) to the surface in 5 days, a
back-of-the-envelope calculation for the corresponding vertical wind speed leads to 2.3
cm/s, which corresponds to the typical synoptic-scale vertical velocity. Therefore, we
look at events which are in line with a "uniform" descent rate consistent with synoptic-
scale weather. Obviously, if the descent takes place in a shorter time, the associated
vertical wind speed would be higher.*

*A new sentence has been added to justify the choice of 5 days (Page 10, Lines 3–
5): "The maximum value for $\Delta t$ was chosen according to the typical lifetime values
for a stratospheric intrusion into the troposphere (see Stohl et al., 2000; Bourqui and
Trépanier, 2010; Trickl et al., 2014, 2016).".*

*Bourqui, M. S., and Trépanier, P.-Y.: Descent of deep stratospheric intrusions during
the IONS August 2006 campaign, J. Geophys. Res., 115, D18301, 2010.*

*Trickl, T., Vogelmann, H., Giehl, H., Scheel, H.-E., Sprenger, M., and Stohl, A.: How
stratospheric are deep stratospheric intrusions?, Atmos. Chem. Phys., 14, 9941-9961,
2014.*

*Trickl, T., Vogelmann, H., Fix, A., Schäfler, A., Wirth, M., Calpini, B., Levrat, G., Ro-
manens, G., Apituley, A., Wilson, K. M., Begbie, R., Reichardt, J., Vömel, H., and
Sprenger, M.: How stratospheric are deep stratospheric intrusions? LUAMI 2008, At-
mos. Chem. Phys., 16, 8791-8815, 2016.*

*5. Links with ENSO, QBO and sunspots poorly convincing*
In Section 4.2.2, the authors claim that some IMFs are correlated with various indica-
tors (of ENSO, QBO, solar activity), but I find that Figure 5 and 6 poorly support these
results (at least when examined by eye). Could these correlations be demonstrated

more clearly, for instance by means of scatterplots?

Beyond this, correlation is not causality. A correlation is interesting to consider only if one suspects some mechanism linking two quantities. In the text, the possible link between ENSO and STE is discussed, but to a much lesser extent the links with the QBO and the solar activity. Could the authors discuss or even speculate a bit more about this?

*According to the Reviewer's comment, we modified the text as follows, to give a deeper indication on the correlations that exist between the parameters: (Page 12, Lines 9–11): "...that is weakly anti-correlated (r = -0.3) to the Quasi-Biennial Oscillation (QBO); the anti-correlation is maximized during post-monsoon and winter seasons (r = -0.5 and r = -0.4, respectively)." And (Page 12, Lines 22–23): "...IMF6 presents some periods of inverse variability with respect to the Multivariate ENSO Index...".*

*Moreover, the mechanisms for which QBO affects stratospheric circulation (and thus STE) are fully explained in Neu et al. (2014) and references therein. To better clarify this part in the manuscript, a new sentence and references have been added to the text (Page 12, Lines 14–17): "More generally, the mechanisms for which QBO affects the STE variability are both the direct modulation of the circulation through thermal wind balance, and the impact on the strength of the overturning circulation by altering the propagation and dissipation of planetary-scale waves, which enhance the meridional circulation and the cross-tropopause transport (Tung and Yang, 1994; Kinnersley and Tung, 1999; Neu et al., 2014).".*

*Furthermore, to better characterize the correlation with the solar activity, a sentence has been added, as well as new references (Page 12, Lines 30–32): "Signals of influence of the sunspot cycle in the upper troposphere-lower stratosphere have been indicated in several works (e.g., Labitzke and Van Loon, 1997a, b; Coughlin and Tung, 2004), suggesting that the association between the Sun and stratospheric parameters (e.g., $O_3$) is due to solar-induced changes in the atmospheric circulation.".*

*Kinnersley, J. S., and Tung, K. K.: Mechanisms for the Extratropical QBO in Circulation*

*and Ozone, J. Atmos. Sci., 56, 1942-1962, 1999.*

*Labitzke, K., and Van Loon, H.: The signal of the 11-year sunspot cycle in the upper troposphere-lower stratosphere, Space Sci. Rev., 80, 393-410, 1997a.*

*Labitzke, K., and Van Loon, H.: Total ozone and the 11-yr sunspot cycle, J. Atmos. Sol.-Terr. Phys., 59, 9-19, 1997b.*

*Tung, K. K., and Yang, H.: Global QBO in circulation and Ozone. Part II: a simple mechanistic model, J. Atmos. Sci., 51, 2708-2721, 1994.*

*6. Balance between paper main body and supplementary material*

The article main body is quite concise in it present form, and I think there is perhaps room for moving important elements from the supplementary material into the paper main body.

For instance, the criteria to detect SIO are of primary importance in the study and could appear in the article, as well as Table S1, and perhaps also Figures S4 and S5.

*We agree with the Reviewer's suggestion of moving relevant part of the Supplementary Material into the main body of the paper. As reported in our response to Reviewer's comment #3, we moved Section S1.4 into Sect. 2. Also Table S1 has been moved into Sect. 4.1, and references to the tables have been updated throughout the text. On the other hand, we decided to keep Figures S4 and S5 into the Supplementary Material.*

**Specific comments**

p.1, l.2: The use of upper-case letters suggests that "STEFLUX" is an acronym. In this case, could the authors make it explicit at least once in the abstract and in the main text body? If it is no acronym but a simple proper noun, I suggest one should write "Steflux".

*STEFLUX stands for "Stratosphere-to-Troposphere Exchange Flux", it has now been mentioned in the Abstract and in the Introduction.*

p.1, l.19: Please consider to change "relating" by "linking".

*This sentence was changed in the Discussion paper version, being: "Furthermore, for the first time, by using the STEFLUX outputs, we investigate the potential impact of specific climate factors (i.e. ENSO, QBO and solar activity) on SI frequency variability over the Mediterranean basin and the Himalayas."*

p.1, l.9-10: "show still"→"still show".

*Done.*

p.2, l.14, "anticyclonic": Do the author mean "cyclonic" instead?

*We mean "anticyclonic", i.e., following the downward transport of air-masses already intruded deeply into the lower troposphere.*

p.2, l.17, "due to anthropogenic emissions": I would specify: local or regional. Please also consider that local or regional biogenic emissions may also alter atmospheric composition with respect to the tropospheric background.

*Done.*

p.2, l.17: "make"→"makes".

*Done.*

p.2, l.27, "Many different methods are based on this combined approach (...) and vary considerably between different measurement sites.": These statements are supported by no literature reference. Could the author cite here a list of references or at least a review paper on the topic? What are those considerable variations between the method? Could the author be a bit more explicit? See also my general comment 1.

*Please refer to our answer to Reviewer's comment #1.*

p.2, l.27, "occurring over": reaching? detected?

*"occurring over" has been changed to "reaching".*

p.2, l.33-34, "Moreover, ...": It seems that this potential application is not illustrated in

the paper. Could the author justify this statement?

*Because the $O_3$ mixing ratio is one of the parameters that are given at every point along the STE climatology trajectories (see Sect. 3.2), STEFLUX could also be deployed for linking $O_3$ concentrations deriving from SI to $O_3$ variations recorded at measurement sites. However, the aim of this paper is to present STEFLUX and compare it to the in situ methodologies deployed at two high-mountain stations, without giving indications on how the SI long-term variability has affected $O_3$ measurements at those sampling sites. Currently, this other potential application of STEFLUX is under study and will probably be part of a future work. We agree with the Reviewer that this statement could be misleading, if placed in the Introduction, thus we moved it in the Conclusions section, where more appropriate (Page 13, Lines 20–22): "Moreover, although not investigated in this work, STEFLUX might be deployed as a particularly relevant tool to investigate how SI long-term variability influences the atmospheric composition at these specific locations (e.g., by deploying the $O_3$ values that are available along each trajectory)".*

p.3, l.5: "to it"→"on climate".

*Done.*

p.3, l.19-20: This statement is questionable and deserves further discussion. See my general comment 2.

*Please refer to our answer to Reviewer's comment #2.*

p.3, l.24-25, "starting at the measurement site": this is too imprecise, especially concerning the altitude. Was the true site altitude or the model surface altitude used to initialize the backtrajectories?

*The starting altitude for NCO-P back-trajectories was 490 hPa, as also reported in Sect. S1.3, to minimize possible effects between the model and the real topography. However, in order to avoid misunderstandings, the sentence "starting at the measurement*

*site" has been removed.*

p.4, whole Section 3.1: even though the case study clarifies well what STEFLUX is (Sect. 3.2), Section 3.1 presenting the tool is confusing. Especially, it is hard to distinguish what comes from Skerlak et al. and what is specific to STEFLUX. Beyond this, a number of elements from Skerlak et al.'s methodology are mentioned in the text (trajectories extended 4 days prior to tropopause crossing; 3D labeling) but it seems these details are not needed in STEFLUX or at least in this paper. If really not needed, these information items are confusing and should be removed. Otherwise, it should be explained why they are important. More generally, I think that the whole Section 3.1 should be rewritten and clarified.

*According to the Reviewer's suggestion, Sect. 3.1 has been rewritten and clarified. Several details characterizing the input STE trajectories have been removed, not to create too much confusion. Additionally, the list of the output files produced by STE-FLUX has been provided (Page 5, Lines 21–24): "STEFLUX produces several output files, which enclose: (i) the trajectory positions and timing found within the box, (ii) the first box crossing positions and timing for each trajectory, (iii) the tropopause crossing position and timing for each trajectory, (iv) the complete list of the trajectories that have crossed the box.".*

p.4: title of Section 3.2 could be changed to "Illustrative case study".

*Done.*

p.4, l.29: The box centered at NCO-P is hardly visible in Fig.1b. Anyway, a reference to this Figure is not useful in this sentence, and mention to Fig.1b could be simply removed here.

*Done.*

p.4, l.30: "recorded" can be removed.

*Done.*

p.5, l.5: in the present form of the paper, the criteria are actually introduced in the supplementary material, not in Section 2. See also my general comment 6.

*In the revised version of the manuscript, the criteria are fully explained in Sect. 2, see our answer to Reviewer's comments #3 and #6.*

p.5, l.11 and ff.: it seems from these lines that there are three different output files from a STEFLUX run, but it is not clear what is in those files. This should be clarified (perhaps in Section 3.1).

*Please refer to our answer above concerning Sect. 3.1.*

p.5, l.17: "indicated in previous studies ..."→"identified as a preferred region for tropopause crossing in previous studies ...".

*Done.*

p.5, l.26, "they still maintained a stratospheric signature": poor expression, please rephrase.

*The sentence has been changed to: "they still followed the stratospheric circulation steered by the subtropical jet stream".*

p.5, l.33: the choice of an horizontal extension of $3° \times 3°$ should be justified briefly.

*The choice of a $3° \times 3°$ horizontal extension was made after performing some sensitivity tests on this parameter. The chosen extension was the one presenting the best agreement with the SIO time series. However, it has to be noted that this parameter can be completely chosen by the user, adapting it to the very different situations (e.g., topography, surrounding regions) of the area under study. The text has been modified as follows (Page 6, Line 32): "...site, after performing a sensitivity test on this parameter (not shown).".*

p.6, l.2-3, "The selected time periods were the same as in Sect.2": please specify.

*Done, a new sentence has been added (Page 7, Lines 1–2): "(i.e., March 2006—December 2013 for NCO-P and January 1998—December 2010 for Mt. Cimone)".*

p.6, l.4: "a table listing ..."→"Table S1 listing ...".

*This sentence has been modified to "Table 1 listing", since Table S1 has been moved into the main body of the text.*

p.6, Section 4.1.1: What is the criterion to tag a day as SI day according to STEFLUX? Is only one box crossing at any moment of the day and of any duration needed? The author should specify this in this Section. (See also the corresponding comment from the Anonymous Referee #1.)

*The criterion to tag a day as SI day is the threshold of at least two box crossings per day, independently on the time. Please see also our comment to Reviewer #1, since we have moved a sentence from Sect. 4.1.2 to Sect. 4.1.1.*

p.6, l.11, "at the two measurement sites": not needed and a bit confusing, please remove.

*Done.*

p.6, l.24: "subtle" is unexpected as adjective for the inter-annual variability. Please rephrase.

*The sentence has been rewritten (Page 7, Line 23): "Although the seasonality was a feature well captured by STEFLUX, the inter-annual variability was less clearly identifiable.".*

p.7, l.1, "criteria coverage": please define. Is it the fraction of time when the data used in the criteria are simultaneously available? Every criterion does not use all the data: what does happen when one data is missing for one criterion but another criterion is fulfilled? Or none other fulfilled? Is the day tagged as SI/non-SI day or discarded? Please clarify.

*As specified in the former Sect S1.4 in the Supplementary Material, and in Sect. 2 of the revised version, a day is selected as "SI day" if at least one criterion is fulfilled. Thus, a specific day can be simultaneously selected by different criteria, but the simultaneity is not strictly required for tagging the day as "SI day". The "criteria coverage" displayed in Fig. 2 is defined as the seasonally averaged percentage of available data from each criterion. The sentence in Page 7, Lines 9–11 has been rewritten to clarify this aspect: "Additionally, the seasonally averaged percentage of available data from each criterion (hereinafter referred to as "criteria coverage") is also reported in the plot (grey bars).".*

p.7, l.22 and ff.: I had a hard time to understand those contingency tables. Considering for instance Table 1(a), does 55 means that during 55 SIO events, STEFLUX detected more than 50% of time of the episode as SI? Does 148 means that during 148 SIO events, STEFLUX detected less than 50% of time of the episode as SI? etc. Please explain a bit more how those numbers should be interpreted. See also the comment from the Anonymous Referee #1 concerning the definitions of accuracy and false alarm rate: how exactly are the presented scores calculated?

*The contingency tables and the related description have been made clearer. We inserted formulas concerning the skill scores we presented, and we gave a description of each parameter composing Table 1. Please also refer to our comments to Reviewer #1.*

p.8, l.2 and 5: the capture rates given in Table 2 (22-27%) are closer to one quarter than to one third.

*Corrected.*

p.8, l.20-24: In case of long travel time and high mixing, can one still consider the air mass as a stratospheric intrusion? See my general comment 4 on stratospheric intrusion lifetime.

*Please refer to our answer to Reviewer's comment #4.*

p.9, l.10: this again is related to my general comment 4: is it really relevant to be irrespective of the degree of mixing and dilution in the troposphere?

*Please refer to our answer to Reviewer's comment #4.*

p.9, l.23-25: could the author explain this statement?

*This sentence has been rewritten as: "First, the location of the crossing is useful to determine the $O_3$ concentration of the air parcels at the start of their tropospheric path towards the target region.".*

p.9, l.32: "If divided seasonally"→"Considering seasons separately"

*Done.*

p.9, l.33 and p.12 l.15: "southward of"→"south of"

*Done.*

p.11, l.12 "does not exhibit as"→"exhibits no"

*Done.*

p.11, l.18 "defined"→"user-defined"

*Done.*

p.11, l.19 "representative"→"illustrative"

*Done.*

p.12, l.17 "both of the"→"both"; "significant"→"statistically significant"

*Done.*

p.13, Appenzeller and Davies, 1992: insufficient reference.

*Corrected.*
p.16, Table 2: missing "(b)".

*Corrected.*

p.18, figure legend, l.3: "Sect. 2"→"Sect. S1.4". See my general comment 6.

*Since the SIO criteria are now fully introduced in Sect. 2, we have not modified this caption.*

p.19, figure 3: the STEFLUX and SIO panel columns could be interchanged, so that the panels (a-d) are numbered in the same order as in the text. Why do the box plots in the upper panel have no whiskers?

*Figure 3 has been redrawn, with the SIO and STEFLUX columns interchanged, as they appear in the text (thus modified accordingly). The box plots for NCO-P have no whiskers (representing the 10th and 90th percentiles), because too few data were available for their calculation.*

Supplementary material
p.1, l.18: do the authors mean gamma-spectroscopy?

*Yes, corrected.*

p.1, l.25: "total column OF ozone".

*Done.*

p.2, l.9-10: This sentence is not fully clear. What does "centered at an ending altitude" mean? Is 490hPa the real altitude of NCO-P? In the same vein in l.12, do the trajectories reach Mt. Cimone at its real altitude level? What is the corresponding pressure level?

*The sentence has been changed to "starting at". The choice of using 490 hPa, higher than the real altitude of NCO-P (5079 m a.s.l., or average pressure of 550 hPa), was made for minimizing possible effects between the model and the real topography. Sim-*

[Figure]

*ilarly, back-trajectories at Mt. Cimone have been started at 2200 m. The text in the Supplementary Material has been changed accordingly.*

p.2, Section S1.4 (SI selection criteria): see my general comments 3 and 6.

*Section S1.4 has been integrated in Sect. 2.*
* * *

---

## Author Response (AR2)

Object: Response to reviewers' comments on "STEFLUX, a tool for investigating stratospheric intrusions: application to two WMO/GAW global stations" by Davide Putero et al.

**Dear Editor,**

Please find below our point-to-point replies (bold italic) to the specific comments raised by the two reviewers. We believe all comments have been addressed and we followed all suggested changes. Modifications as respect to the original manuscript are included in the new version uploaded, and marked with red and blue colors in the manuscript version below.

We thank the reviewers for their useful and valuable comments and we hope the manuscript now meets the journal's specific standards for publication.

Yours sincerely,

Davide Putero (on behalf of all the co-authors)

**Reviewer #1:**

General Comments: This paper presents STEFLUX, a new tool that detects stratospheric intrusions affecting a specific location during a specific time period. STEFLUX is well described and the results are thoroughly discussed revealing the benefits and restrictions of the tool. As the transport of stratospheric air masses into the troposphere is of great importance, STEFLUX can be used in conjunction with observations for several scientific purposes. Therefore, I consider the paper to be an interesting study and recommend its publication in ACP, but only after addressing the following comments.

We thank the Reviewer for his/her valuable suggestions and his/her encouraging evaluation. In the following, we report our point-to-point replies to each of the raised points. Modifications to the text are performed in the revised version of the manuscript and are marked in red and blue colors in the manuscript version below.

Main comments: It seems that there are inconsistencies between the skill scores values (False Alarm Rate, ORSS) presented in the manuscript (Page 7 line 27 – Page 8 line 5, Table 1) and the contingency tables presented in Table 1. Moreover, the presentation of the results in Table 1 needs to be more reader-friendly. I suggest the following:

1. Include (in Section 4.1.2) the formulas used to calculate all skill scores along with the corresponding references, i.e. ORSS=(AD-BC)/(AD+BC) (Thornes and Stephenson, 2001), explaining what A, B, C and D stands for in your case.

2. Assign A, B, C and D to the respective values in Table 1.

3. Check calculations for the skill scores. It is likely that your results are better (higher ORSS values and lower False Alarm Rate values).

4. Add a label for each table in Table 1, in order to be clear which approach is the "reference" and which is the "predictor". i.e for Table 1a,c "SIO vs STEFLUX" and for Table 1b,d "STEFLUX vs SIO".

5. Revise the discussion (for skill scores) in the manuscript if needed.

We thank the Reviewer for this important comment, and we apologize for the inconsistencies between the wrong values given in the text and those reported in Table 1. Our new results are indeed better than those presented in the first version of the manuscript, with a lower false alarm rate (hereinafter called Probability of False Detection, POFD, equal to 0.45), and higher ORSS (see the updated Table 1) for all comparisons. These values have been updated in the text (Page 8, Lines 24-25) and in Table 1.

Moreover, as suggested by the Reviewer, we made the description of the skill scores and the layout of Table 1 clearer. Formulas for the accuracy, false alarm ratio, probability of false detection and ORSS have been inserted (following Thornes and Stephenson, 2001), explaining also what A, B, C and D stand for (see Sect. 4.1.2). Table 1 has been updated by specifying what these capital letters refer to and by adding labels for the "STEFLUX vs SIO" or "SIO vs STEFLUX" comparisons. The caption has also been modified accordingly.

Minor comments: Please add degree symbols for lon and lat values in the manuscript. i.e. Page 3, line 20.

Done.

Section 4.1.1: Please include a definition for STEFLUX SI day. i.e. threshold of at least 1 box crossing per day?

The definition of SI day was erroneously given in Sect. 4.1.2 (Page 7, Lines 10-11): "...SI days (with a threshold of at least 2 box crossings per day for STEFLUX, in order to retain robust information only and to discharge "erratic" events).". Thus, this sentence has been moved above (Page 7, Lines 17-18).

Page 7, lines 6-7: "(see the Supplementary Material)". Please specify exactly where in the Supplementary Material.

Done. Since old Table S1 has been moved into the main body of the text (see our answer to Reviewer #2 comments), the sentence has been modified to: "(see Table S1 in the Supplementary Material)".

Table 2: Add "(b)" in the second table. *Done*.

Figure 1a: Please replace "STEFLUX [#]" with "STEFLUX [number of crossings]". *Done.*

Figure 4: The map continents are not so clear. Please change map continents color if possible (maybe grey).

Done.

**Reviewer #2:**

In this study, the authors present a new tool, called STEFLUX, to select trajectories having crossed the tropopause downward at some time in the past days and arriving into a user-defined geographical box within a prescribed time-window. The trajectories are selected among a large set of pre-computed trajectories based on the ERA-Interim reanalysis from the ECMWF. Doing this, this is presumably a fast-computing tool since no trajectory computation is needed. Output data allow for various applications, such as assessing the occurrence frequency of stratospheric intrusions (SI) in the lower troposphere at any place on Earth at regional scale, but also characterizing preferred entry regions in the UTLS, travel times until the target area, etc. The paper presents an illustrative case study, a skill assessment study with respect to SI detection based on (mainly ground-based) observations, and finally a climatology over 35 years of SI events over two focal areas.

STEFLUX is certainly a promising tool which may be helpful for a scientific community larger than the authors' research team. The paper itself is fairly well-built and written, and the presented scientific material and discussions are of good scientific quality.

Therefore, I recommend the publication of this study in ACP, but not before the author take in consideration the following comments and propose a revised version of their manuscript. I would appreciate if the authors could pay particular attention to my general comments 3 and 4.

We thank the Reviewer for his/her valuable suggestions and his/her encouraging evaluation. In the following, we report our point-to-point replies to each of the raised points. Modifications to the text are performed in the revised version of the manuscript and are marked in red and blue colors in the manuscript version below.

**General comments**

**1. Method originality not fully clear**

While reading Section 1, it is not straightforward to know what is new in the STEFLUX method compared to existing methods based on backtrajectories. For instance, one could wonder why don't the author simply initialize backtrajectories from the target regions and see if their cross the tropopause at some time in the past?

I guess one major advantage of the method is computation speed, and this is due to the fact that it works from pre-computed backtrajectories. But this is not clearly stated in the text. More generally, I think the Introduction could be developed and depict more explicitely the state-of-the-art in the domain: what are the different types approaches? what are their drawbacks or limitations? etc. The originality of the STEFLUX method should thus be more emphasized. *As correctly pointed out by the Reviewer, one characteristic of STEFLUX is the computational speed, since no calculation of back-trajectories is required. This would be a particularly long and time-consuming task, especially when evaluating STE over long periods. The calculation of backward trajectories from the target point would have been possible for NCO-P and Mt. Cimone, but the aim of STEFLUX is to be a tool that can be easily applied at any point on the globe. Therefore, it is valuable to have a set of pre-computed forward trajectories (please refer to Škerlak et al., 2014, to see how the trajectories are generated), because a simple calculation of backward trajectories is not feasible. Also, as demonstrated in the paper, it is sufficient specifying only a few parameters (spatial coordinates and top lid of the box) in STEFLUX, for obtaining a quick and reliable estimate of the STE occurred over the desired time window. In addition to this,*  being based on the constantly updated ERA-Interim reanalysis, STEFLUX allows for a STE estimate back to 1979, thus it is especially important from a climatological perspective. In order to highlight these motivations in the text, we modified as follows (Page 3, Lines 7-8): "The tool, called STEFLUX (Stratosphere-to-Troposphere Exchange Flux), is a relatively fast-computing algorithm which makes use of the pre-computed trajectories composing the STE climatology by..." and (Page 3, Lines 12-14): "Its computational speed and user-friendly approach (it is sufficient to specify only a few parameters to work) make it suitable for obtaining a quick and reliable estimate of the SI occurred at a specific place over the desired time window (including long periods which would otherwise require a lot of time-consuming calculations). A potential...".

Moreover, the Introduction part concerning the different "observations-based" methodologies to detect STE has been developed including several works, together with one review paper (Pages 2-3): "Usually, stratospheric influence is detected at a measurement site by analyzing the variability of in situ "stratospheric" observations (e.g., relative humidity, 7Be, 10Be, O3, atmospheric pressure variability) and profiling datasets (radio/ozone-sondes), coupled with the analysis of satellite (e.g., total column of ozone) and various kinds of numerical weather prediction (NWP) model products fields. Many different methods, as thoroughly reviewed in Stohl et al. (2003), are based on this combined approach. Stohl et al. (2000) deployed a detection algorithm based on the in situ variation of experimental data and simulations with a passive stratospheric tracer. Similarly, other studies analyzed STE by coupling experimental data and back-trajectories (e.g., Cristofanelli et al., 2006, 2010; Trickl et al., 2010). Usually, specific threshold values are applied to in situ tracers' variability to detect the presence of air-masses with stratospheric "fingerprints". Also trajectory and dispersion models are extensively used to detect the occurrence of STE. For example, Cui et al. (2009) used the particle dispersion model FLEXPART (Stohl et al., 2005) and the trajectory model LAGRANTO (Wernli and Davies, 1997) to identify stratospheric transport at the high-altitude Alpine site Jungfraujoch (Switzerland), while Tarasova et al. (2009) deployed 3D air-mass back-trajectories to trace the atmospheric transport at two high mountain measurement sites over the Alps and Caucasus. As pointed out by Bourqui (2006), trajectory-based approaches can provide a lower-bound estimate for STE flux, while dispersion models can provide slightly larger estimates. Typically, when used to detect STE at specific locations at the Earth's surface, all of these "observations-based" methodologies vary among different measurement sites, with respect to the number and types of stratospheric tracers available/considered, threshold values adopted, and often require a lot of time-consuming implementation to work. Moreover, it should be argued that none of the most diffused tracers have a "pure" stratospheric origin; for example, 7Be and  $O_3$  are affected by significant tropospheric sources. Furthermore, the compilation of proper long-term climatologies is very often hindered by the lack of long-term observations of "stratospheric" tracers.".

Stohl, A., Forster, C., Frank, A., Seibert, P., and Wotawa, G.: Technical note: The Lagrangian particle dispersion model FLEXPART version 6.2, Atmos. Chem. Phys., 5, 2461-2474, 2005. Tarasova, O. A., Senik, I. A., Sosonkin, M. G., Cui, J., Staehelin, J., and Prévôt, A. S. H.: Surface ozone at the Caucasian site Kislovodsk high mountain station and the Swiss Alpine site Jungfraujoch: data analysis and trends (1990–2006). Atmos. Chem. Phys., 9, 4157-4175, 2009.

2. Representativity of a deep valley station

At several places in the text, it is suggested (e.g. when mentioning the "overpass" effect) that SIO may be missed at the surface stations because their measurement may not always be representative of the free troposphere at regional scale owing to local mountain meteorology. I think this concern is especially true for the NCO-P station, which is located in the bottom of a deep valley. Even in conditions of down-valley flow, it is likely that air has been in contact with the surface before reaching the observatory. Ozone in particular may have experienced deposition, and surface ozone concentrations may be lower than those encountered in the free troposphere. Valleys are indeed known to be net sinks for ozone (see e.g. Furger et al., Atmos. Env., 34, 1395-1412; Wotawa and Kromp-Kolb, Atmos. Env., 34, 1319-1322).

Even in the cited references (Bonasoni et al., 2010; Cristofanelli et al. 2010) little is said on the station representativeness at regional scale (except in the monsoon season at night). It would be worthy if this question could be briefly discussed somewhere in the paper (e.g. in Section 2 when the station are presented).

In contrast, I am much more confident in the regional representativeness of the mountain-top site Monte-Cimone (of course, apart from anabatic conditions) and the suitability of the site to detect deep stratospheric intrusion, although it is at much lower altitude.

The Reviewer raised an interesting point concerning the representativeness of NCO-P ozone observations. Unfortunately, no specific studies aimed at assessing these kinds of processes at NCO-P have been carried out so far. For this reason, in Sect. 4.1.3, the following sentences have been added, as well as two new references (Page 10, Lines 15-19): "Furthermore, for NCO-P, it should be considered that the station is located in a narrow valley. Thus, it is conceivable that, during the transport within the valley,  $O_3$  (one of the stratospheric tracers considered by SIO) experiences deposition phenomena, thus decreasing the actual concentration that the stratospheric air-mass would have in the free troposphere (see, e.g., Furger et al., 2000; Wotawa and Kromp-Kolb, 2000).".

Furger, M., Dommen, J., Graber, W. K., Poggio, L., Prévôt, A. S., Emeis, S., Grell, G., Trickl, T., Gomiscek, B., Neininger, B., and Wotawa, G.: The VOTALP Mesolcina Valley Campaign 1996 – concept, background and some highlights, Atmos. Environ., 34, 1395-1412, 2000. Wotawa, G., and Kromp-Kolb, H.: The research project VOTALP – general objectives and main results, Atmos. Environ., 34, 1319-1322, 2000.

**3. SIO detection criteria too imprecise**

In Section S1.4 (supplementary material), the SIO selection criteria are presented in a too vague and qualitative manner (and therefore the criteria appear to be subjective). For instance, what does "significant variation of daily P" mean? What is the threshold to consider the variation is significant? Further, is the current pressure daily mean compared to the value the day before? One could ask such questions for almost every items of the two lists. The authors must present their study in a reproducible way, and those criteria are central elements. This section should be rewritten in a much more rigourous and quantitative manner, with the concern of study reproducibility. *We apologize for having provided too little detail when describing the SIO criteria. The significant variations of the several parameters (TCO, P, 7Be) are computed by the following methodology: first, a three-time repeated iteration of a 21-days running mean (the so-called Kolmogorov-Zurbenko filter, see Sebald et al., 2000) is applied to the daily average time series; then, residuals are computed by subtracting these latest values from the daily averages; residuals*

which exceed the upper (for 7Be) or upper and lower (for TCO and P) endpoints of the 95% confidence interval of the residuals distribution over the whole period are thus labelled as "significant variations".

In order to make the SIO selection methodology clearer and available in the main text, Sect. S1.4 has been rewritten including these detailed information and moved to Sect. 2.

**Sebald, L., Treffeisen, R., Reimer, E., and Hies, T.: Spectral analysis of air pollutants. Part 2: ozone time series, Atmos. Environ., 34, 3503-3509, 2000.**

**4. Missing discussion on backtrajectory maximum duration**

In this study, tropopause crossings are considered up to 5 (= 1+4) days prior the trajectories reach the target box. But if one goes sufficiently deep backward in time, any trajectory ending in the target box crossed the tropopause at some time in the past. On the contrary, if the trajectory maximum duration is reduced below some value, no SI at all is detected.

Actually, the target region can be found to be from 0% to 100% of the time under the influence of stratospheric intrusions, depending on the chosen trajectory maximum duration. This parameter appears to be of central importance in the STEFLUX tool. I think a sensitivity study to this parameter should be presented (especially in relation with the results (percentages) given in Section 4.2.1), or at least, the choice of 5 days (which obviously comes from the work by Skerlak et al., 2014) should be carefully discussed and justified.

This leads to a more general question: any sufficiently long-lived molecule in the troposphere resided in the stratosphere at some prior times. What is the typical lifetime of a stratospheric intrusion in the troposphere, and when should one consider the air mass composition as being no longer influenced by the stratosphere?

I think these points are crucial in this study and deserve thorough discussions.

The time aspect is always a difficult task in trajectory analyses. The typical lifetime of a stratospheric intrusion in the troposphere has been considered in several papers (e.g., Stohl et al., 2000; Bourqui and Trépanier, 2010; Trickl et al., 2014, 2016). Stohl et al. (2000) argue that, once brought into the troposphere, the stratospheric signature of an air-mass (i.e., low RH, high  $O_3$ ) gets lost over the period of a few days, because it gets quasi-adiabatically stirred by large-scale cyclonic and anticyclonic disturbances. The choice of 5 days for trajectories is also presented in Bourqui and Trépanier (2010). It was found that the trajectory clusters for their case studies experienced three distinct phases during their descent from the stratosphere (namely crossing of the tropopause, free descent, and quasi-horizontal dispersion in the troposphere), and this whole process takes typically 4-5 days. Similarly, Trickl et al. (2016) define "stratospheric intrusion trajectories" as those initially residing in the stratosphere and descending during the following 5 days by more than 300 hPa into the troposphere. Finally, we could justify our choice with a simple order-of-magnitude calculation. If we suppose that an air parcel descends uniformly from the tropopause (~10 km) to the surface in 5 days, a back-of-the-envelope calculation for the corresponding vertical wind speed leads to 2.3 cm/s, which corresponds to the typical synopticscale vertical velocity. Therefore, we look at events which are in line with a "uniform" descent rate consistent with synoptic-scale weather. Obviously, if the descent takes place in a shorter time, the associated vertical wind speed would be higher.

A new sentence has been added to justify the choice of 5 days (Page 10, Lines 3-5): "The maximum value for  $\Delta t$  was chosen according to the typical lifetime values for a stratospheric

*intrusion into the troposphere (see Stohl et al., 2000; Bourqui and Trépanier, 2010; Trickl et al., 2014, 2016)."*.

Bourqui, M. S., and Trépanier, P.-Y.: Descent of deep stratospheric intrusions during the IONS August 2006 campaign, J. Geophys. Res., 115, D18301, 2010.

Trickl, T., Vogelmann, H., Giehl, H., Scheel, H.-E., Sprenger, M., and Stohl, A.: How stratospheric are deep stratospheric intrusions?, Atmos. Chem. Phys., 14, 9941-9961, 2014. Trickl, T., Vogelmann, H., Fix, A., Schäfler, A., Wirth, M., Calpini, B., Levrat, G., Romanens, G., Apituley, A., Wilson, K. M., Begbie, R., Reichardt, J., Vömel, H., and Sprenger, M.: How stratospheric are deep stratospheric intrusions? LUAMI 2008, Atmos. Chem. Phys., 16, 8791-8815, 2016.

**5. Links with ENSO, QBO and sunspots poorly convincing**

In Section 4.2.2, the authors claim that some IMFs are correlated with various indicators (of ENSO, QBO, solar activity), but I find that Figure 5 and 6 poorly support these results (at least when examined by eye). Could these correlations be demonstrated more clearly, for instance by means of scatterplots?

Beyond this, correlation is not causality. A correlation is interesting to consider only if one suspects some mechanism linking two quantities. In the text, the possible link between ENSO and STE is discussed, but to a much lesser extent the links with the QBO and the solar activity. Could the authors discuss or even speculate a bit more about this?

According to the Reviewer's comment, we modified the text as follows, to give a deeper indication on the correlations that exist between the parameters: (Page 12, Lines 9-11): "...that is weakly anti-correlated (r = -0.3) to the Quasi-Biennial Oscillation (QBO); the anti-correlation is maximized during post-monsoon and winter seasons (r = -0.5 and r = -0.4, respectively)." And (Page 12, Lines 22-23): "...IMF6 presents some periods of inverse variability with respect to the Multivariate ENSO Index...".

Moreover, the mechanisms for which QBO affects stratospheric circulation (and thus STE) are fully explained in Neu et al. (2014) and references therein. To better clarify this part in the manuscript, a new sentence and references have been added to the text (Page 12, Lines 14-17): "More generally, the mechanisms for which QBO affects the STE variability are both the direct modulation of the circulation through thermal wind balance, and the impact on the strength of the overturning circulation by altering the propagation and dissipation of planetary-scale waves, which enhance the meridional circulation and the cross-tropopause transport (Tung and Yang, 1994; Kinnersley and Tung, 1999; Neu et al., 2014).".

Furthermore, to better characterize the correlation with the solar activity, a sentence has been added, as well as new references (Page 12, Lines 30-32): "Signals of influence of the sunspot cycle in the upper troposphere-lower stratosphere have been indicated in several works (e.g., Labitzke and Van Loon, 1997a, b; Coughlin and Tung, 2004), suggesting that the association between the Sun and stratospheric parameters (e.g.,  $O_3$ ) is due to solar-induced changes in the atmospheric circulation.".

Kinnersley, J. S., and Tung, K. K.: Mechanisms for the Extratropical QBO in Circulation and Ozone, J. Atmos. Sci., 56, 1942-1962, 1999.

Labitzke, K., and Van Loon, H.: The signal of the 11-year sunspot cycle in the upper troposphere-lower stratosphere, Space Sci. Rev., 80, 393-410, 1997a. Labitzke, K., and Van Loon, H.: Total ozone and the 11-yr sunspot cycle, J. Atmos. Sol.-Terr. Phys., 59, 9-19, 1997b.

Tung, K. K., and Yang, H.: Global QBO in circulation and Ozone. Part II: a simple mechanistic model, J. Atmos. Sci., 51, 2708-2721, 1994.

6. Balance between paper main body and supplementary material

The article main body is quite concise in it present form, and I think there is perhaps room for moving important elements from the supplementary material into the paper main body. For instance, the criteria to detect SIO are of primary importance in the study and could appear in the article, as well as Table S1, and perhaps also Figures S4 and S5.

We agree with the Reviewer's suggestion of moving relevant part of the Supplementary Material into the main body of the paper. As reported in our response to Reviewer's comment #3, we moved Section S1.4 into Sect. 2. Also Table S1 has been moved into Sect. 4.1, and references to the tables have been updated throughout the text. On the other hand, we decided to keep Figures S4 and S5 into the Supplementary Material.

Specific comments

p.1, 1.2: The use of upper-case letters suggests that "STEFLUX" is an acronym. In this case, could the authors make it explicit at least once in the abstract and in the main text body? If it is no acronym but a simple proper noun, I suggest one should write "Steflux".

STEFLUX stands for "Stratosphere-to-Troposphere Exchange Flux", it has now been mentioned in the Abstract and in the Introduction.

p.1, l.19: Please consider to change "relating" by "linking".

This sentence was changed in the Discussion paper version, being: "Furthermore, for the first time, by using the STEFLUX outputs, we investigate the potential impact of specific climate factors (i.e. ENSO, QBO and solar activity) on SI frequency variability over the Mediterranean basin and the Himalayas."

p.1, 1.9-10: "show still" $\rightarrow$ "still show". **Done.**

p.2, 1.14, "anticyclonic": Do the author mean "cyclonic" instead? We mean "anticyclonic", i.e., following the downward transport of air-masses already intruded deeply into the lower troposphere.

p.2, l.17, "due to anthropogenic emissions": I would specify: local or regional. Please also consider that local or regional biogenic emissions may also alter atmospheric composition with respect to the tropospheric background.

Done.

p.2, 1.17: "make"  $\rightarrow$  "makes". **Done.**

p.2, 1.27, "Many different methods are based on this combined approach (...) and vary considerably between different measurement sites.": These statements are supported by no literature reference. Could the author cite here a list of references or at least a review paper on the topic? What are those considerable variations between the method? Could the author be a bit more explicit? See also my general comment 1.

Please refer to our answer to Reviewer's comment #1.

p.2, 1.27, "occurring over": reaching? detected? *"occurring over" has been changed to "reaching"*.

p.2, 1.33-34, "Moreover, ...": It seems that this potential application is not illustrated in the paper. Could the author justify this statement?

Because the  $O_3$  mixing ratio is one of the parameters that are given at every point along the STE climatology trajectories (see Sect. 3.2), STEFLUX could also be deployed for linking  $O_3$  concentrations deriving from SI to  $O_3$  variations recorded at measurement sites. However, the aim of this paper is to present STEFLUX and compare it to the in situ methodologies deployed at two high-mountain stations, without giving indications on how the SI long-term variability has affected  $O_3$  measurements at those sampling sites. Currently, this other potential application of STEFLUX is under study and will probably be part of a future work. We agree with the Reviewer that this statement could be misleading, if placed in the Introduction, thus we moved it in the Conclusions section, where more appropriate (Page 13, Lines 20-22): "Moreover, although not investigated in this work, STEFLUX might be deployed as a particularly relevant tool to investigate how SI long-term variability influences the atmospheric composition at these specific locations (e.g., by deploying the  $O_3$  values that are available along each trajectory)".

p.3, 1.5: "to it" $\rightarrow$ "on climate". **Done.**

p.3, 1.19-20: This statement is questionable and deserves further discussion. See my general comment 2.

Please refer to our answer to Reviewer's comment #2.

p.3, 1.24-25, "starting at the measurement site": this is too imprecise, especially concerning the altitude. Was the true site altitude or the model surface altitude used to initialize the backtrajectories?

The starting altitude for NCO-P back-trajectories was 490 hPa, as also reported in Sect. S1.3, to minimize possible effects between the model and the real topography. However, in order to avoid misunderstandings, the sentence "starting at the measurement site" has been removed.

p.4, whole Section 3.1: even though the case study clarifies well what STEFLUX is (Sect. 3.2), Section 3.1 presenting the tool is confusing. Especially, it is hard to distinguish what comes from Skerlak et al. and what is specific to STEFLUX. Beyond this, a number of elements from Skerlak et al.'s methodology are mentioned in the text (trajectories extended 4 days prior to tropopause crossing; 3D labeling) but it seems these details are not needed in STEFLUX or at least in this paper. If really not needed, these information items are confusing and should be removed. Otherwise, it should be explained why they are important. More generally, I think that the whole Section 3.1 should be rewritten and clarified.

According to the Reviewer's suggestion, Sect. 3.1 has been rewritten and clarified. Several details characterizing the input STE trajectories have been removed, not to create too much confusion. Additionally, the list of the output files produced by STEFLUX has been provided (Page 5, Lines 21-24): "STEFLUX produces several output files, which enclose: (i) the trajectory positions and timing found within the box, (ii) the first box crossing positions and timing for each trajectory, (iii) the tropopause crossing position and timing for each trajectory, (iv) the complete list of the trajectories that have crossed the box."

p.4: title of Section 3.2 could be changed to "Illustrative case study". *Done.*

p.4, 1.29: The box centered at NCO-P is hardly visible in Fig.1b. Anyway, a reference to this Figure is not useful in this sentence, and mention to Fig.1b could be simply removed here. *Done.*

p.4, 1.30: "recorded" can be removed. *Done*.

p.5, 1.5: in the present form of the paper, the criteria are actually introduced in the supplementary material, not in Section 2. See also my general comment 6.

In the revised version of the manuscript, the criteria are fully explained in Sect. 2, see our answer to Reviewer's comments #3 and #6.

p.5, 1.11 and ff.: it seems from these lines that there are three different output files from a STEFLUX run, but it is not clear what is in those files. This should be clarified (perhaps in Section 3.1).

Please refer to our answer above concerning Sect. 3.1.

p.5, 1.17: "indicated in previous studies ..." $\rightarrow$ "identified as a preferred region for tropopause crossing in previous studies ...". Done.

p.5, 1.26, "they still maintained a stratospheric signature": poor expression, please rephrase. *The sentence has been changed to: "they still followed the stratospheric circulation steered by the subtropical jet stream".*

p.5, 1.33: the choice of an horizontal extension of  $3^{\circ} \times 3^{\circ}$  should be justified briefly. The choice of a  $3^{\circ}x3^{\circ}$  horizontal extension was made after performing some sensitivity tests on this parameter. The chosen extension was the one presenting the best agreement with the SIO time series. However, it has to be noted that this parameter can be completely chosen by the user, adapting it to the very different situations (e.g., topography, surrounding regions) of the area under study. The text has been modified as follows (Page 6, Line 32): "...site, after performing a sensitivity test on this parameter (not shown).".

p.6, 1.2-3, "The selected time periods were the same as in Sect.2": please specify. Done, a new sentence has been added (Page 7, Lines 1-2): "(i.e., March 2006–December 2013 for NCO-P and January 1998–December 2010 for Mt. Cimone)".

p.6, l.4: "a table listing ..." → "Table S1 listing ...".

This sentence has been modified to "Table 1 listing", since Table S1 has been moved into the main body of the text.

p.6, Section 4.1.1: What is the criterion to tag a day as SI day according to STEFLUX? Is only one box crossing at any moment of the day and of any duration needed? The author should specify this in this Section. (See also the corresponding comment from the Anonymous Referee #1.) *The criterion to tag a day as SI day is the threshold of at least two box crossings per day, independently on the time. Please see also our comment to Reviewer #1, since we have moved a sentence from Sect. 4.1.2 to Sect. 4.1.1.*

p.6, l.11, "at the two measurement sites": not needed and a bit confusing, please remove. *Done*.

p.6, 1.24: "subtle" is unexpected as adjective for the inter-annual variability. Please rephrase. *The sentence has been rewritten (Page 7, Line 23): "Although the seasonality was a feature well captured by STEFLUX, the inter-annual variability was less clearly identifiable."*.

p.7, l.1, "criteria coverage": please define. Is it the fraction of time when the data used in the criteria are simultaneously available? Every criterion does not use all the data: what does happen when one data is missing for one criterion but another criterion is fulfilled? Or none other fulfilled? Is the day tagged as SI/non-SI day or discarded? Please clarify.

As specified in the former Sect S1.4 in the Supplementary Material, and in Sect. 2 of the revised version, a day is selected as "SI day" if at least one criterion is fulfilled. Thus, a specific day can be simultaneously selected by different criteria, but the simultaneity is not strictly required for tagging the day as "SI day". The "criteria coverage" displayed in Fig. 2 is defined as the seasonally averaged percentage of available data from each criterion. The sentence in Page 7, Lines 9-11 has been rewritten to clarify this aspect: "Additionally, the seasonally averaged percentage of available data from each criterine to as "criteria coverage") is also reported in the plot (grey bars)."

p.7, 1.22 and ff.: I had a hard time to understand those contingency tables. Considering for instance Table 1(a), does 55 means that during 55 SIO events, STEFLUX detected more than 50% of time of the episode as SI? Does 148 means that during 148 SIO events, STEFLUX detected less than 50% of time of the episode as SI? etc. Please explain a bit more how those numbers should be interpreted. See also the comment from the Anonymous Referee #1 concerning the definitions of accuracy and false alarm rate: how exactly are the presented scores calculated?

The contingency tables and the related description have been made clearer. We inserted formulas concerning the skill scores we presented, and we gave a description of each parameter composing Table 1. Please also refer to our comments to Reviewer #1.

p.8, 1.2 and 5: the capture rates given in Table 2 (22-27%) are closer to one quarter than to one third.

Corrected.

p.8, 1.20-24: In case of long travel time and high mixing, can one still consider the air mass as a stratospheric intrusion? See my general comment 4 on stratospheric intrusion lifetime. *Please refer to our answer to Reviewer's comment #4.*

p.9, 1.10: this again is related to my general comment 4: is it really relevant to be irrespective of the degree of mixing and dilution in the troposphere? *Please refer to our answer to Reviewer's comment #4.*

p.9, 1.23-25: could the author explain this statement?

This sentence has been rewritten as: "First, the location of the crossing is useful to determine the  $O_3$  concentration of the air parcels at the start of their tropospheric path towards the target region.".

p.9, 1.32: "If divided seasonally"→"Considering seasons separately" *Done*.

p.9, 1.33 and p.12 l.15: "southward of" $\rightarrow$ "south of" *Done*.

p.11, 1.12 "does not exhibit as"  $\rightarrow$  "exhibits no" **Done.**

p.11, l.18 "defined"→"user-defined" **Done.**

p.11, l.19 "representative"→"illustrative" **Done.**

p.12, l.17 "both of the"  $\rightarrow$  "both"; "significant"  $\rightarrow$  "statistically significant" *Done.*

p.13, Appenzeller and Davies, 1992: insufficient reference. *Corrected.*

p.16, Table 2: missing "(b)". *Corrected.*

p.18, figure legend, 1.3: "Sect. 2" $\rightarrow$ "Sect. S1.4". See my general comment 6. Since the SIO criteria are now fully introduced in Sect. 2, we have not modified this caption.

p.19, figure 3: the STEFLUX and SIO panel columns could be interchanged, so that the panels (a-d) are numbered in the same order as in the text. Why do the box plots in the upper panel have no whiskers?

Figure 3 has been redrawn, with the SIO and STEFLUX columns interchanged, as they appear in the text (thus modified accordingly). The box plots for NCO-P have no whiskers (representing the 10th and 90th percentiles), because too few data were available for their calculation.

Supplementary material p.1, 1.18: do the authors mean gamma-spectroscopy? *Yes, corrected.*

p.1, l.25: "total column OF ozone". *Done.*

p.2, 1.9-10: This sentence is not fully clear. What does "centered at an ending altitude" mean? Is 490hPa the real altitude of NCO-P? In the same vein in 1.12, do the trajectories reach Mt. Cimone at its real altitude level? What is the corresponding pressure level?

The sentence has been changed to "starting at". The choice of using 490 hPa, higher than the real altitude of NCO-P (5079 m a.s.l., or average pressure of 550 hPa), was made for minimizing possible effects between the model and the real topography. Similarly, back-trajectories at Mt. Cimone have been started at 2200 m. The text in the Supplementary Material has been changed accordingly.

p.2, Section S1.4 (SI selection criteria): see my general comments 3 and 6. *Section S1.4 has been integrated in Sect. 2.*

[revised manuscript text omitted]
    | $26^{\circ} \underbrace{N, 29^{\circ} N}{\infty}$              | $\underbrace{43^{\circ}}_{\times}$ $\underbrace{N, 46^{\circ}}_{\times}$ $\underbrace{N}_{\times}$ |
| Lon_min, Lon_max    | $\underbrace{85^{\circ}}_{E,88} \underbrace{E,88^{\circ}}_{E}$ | $ \stackrel{9^{\circ}}{\underbrace{E, 12^{\circ}}} \stackrel{E}{\underbrace{E}} $                  |
| Box_top             | 550 hPa                                                 | 790 hPa                                                                                     |
| Time span    | 01 Mar. 2006–31 Dec. 2013                               | 01 Jan. 1998–31 Dec. 2010                                                                   |
| Temporal resolution | 1.h                                                     | 1.h                                                                                         |

**Table 2.** 2×2 contingency tables for the comparisons of the SI events time series, i.e., identified by the SIO and STEFLUX approaches, for NCO-P (a, b) and Mt. Cimone (c, d). For each table the Odds Ratio Skill Score (ORSS) is also reported, along with the minimum ORSS required to have real skill at the 99% confidence level (in parentheses, see Thornes and Stephenson, 2001). Capital letters are defined as follows: A indicates the number of SI events selected by both methodologies (STEFLUX and SIO); B represents the number of events selected as SI by the first methodology but as no-SI by the second one; C represents the number of events selected as no-SI by the first methodology but as SI by the second one; and D represents the number of no-SI events selected by both methodologies.

|            |                    |                                            | Ν                                                                               | ICO-P                    |                    |                                                        |                                    |  |
|------------|--------------------|--------------------------------------------|---------------------------------------------------------------------------------|--------------------------|--------------------|--------------------------------------------------------|------------------------------------|--|
| (a)        | "SIO vs STEFLUX"   |                                            |                                                                                 | (b)                      | "STEFLUX vs SIO"   |                                                        |                                    |  |
|            | STEFLUX            |                                            |                                                                                 |                          |                    | SIO                                                    |                                    |  |
|            |                    | SI                                         | no-SI                                                                           |                          |                    | SI                                                     | no-SI                              |  |
| 610        | SI                 | A = 55                              | B = 148                                                           | STEFLUX                  | SI                 | A = 39                                          | B = 116              |  |
| 310        | no-SI              | C = 23                              | D = 181                                                                  |                          | no-SI              | C = 16                                          | D = 140                     |  |
| ORSS       | 0.49 (0.35)        |                                            |                                                                                 | ORSS                     | 0.49 (0.35)        |                                                        |                                    |  |
| Mt. Cimone |                    |                                            |                                                                                 |                          |                    |                                                        |                                    |  |
|            |                    |                                            | Mt.                                                                             | Cimone                   |                    |                                                        |                                    |  |
| (c)        | "SI                | O vs STEF                                  | Mt.
FLUX"                                                                    | Cimone
(d)            | "ST                | EFLUX v                                                | s SIO"                             |  |
| (c)        | "SI                | O vs STEF                                  | Mt.
FLUX"
FLUX                                                            | Cimone
(d)            | "ST                | EFLUX v                                                | s SIO"
SIO                      |  |
| (c)        | "SI                | O vs STEF
STE                           | Mt.
FLUX"
FLUX
no-SI                                                   | Cimone
(d)            | "ST                | EFLUX v
S                                           | s SIO"
HO
no-SI              |  |
| (c)
510 | "SI
SI          | $\frac{O \text{ vs STEF}}{STE}$            | Mt.
FLUX"
FLUX
no-SI
$\underline{B} = 226$                          | Cimone
(d)            | "ST
SI          | $\frac{\text{TEFLUX v}}{\text{SI}}$                    | s SIO"
NO
B = 185            |  |
| (c)
SIO | "SI
SI
no-SI | O vs STEP
STE
SI
A = 73
C = 25 | Mt.
FLUX"
FLUX
no-SI
$\underline{B} = 226$
$\underline{D} = 275$ | Cimone
(d)
STEFLUX | "ST
SI
no-SI | $\frac{\text{EFLUX v}}{\text{S}}$ SI $A = 52$ $C = 13$ | s SIO"
NO
B = 185
D = 225 |  |

| (a)                                                          | NCO-P                                                 |                                                                               | Mt. Cimone                                                  |                                                                               |  |
|--------------------------------------------------------------|-------------------------------------------------------|-------------------------------------------------------------------------------|-------------------------------------------------------------|-------------------------------------------------------------------------------|--|
| SI event duration                                            | SI events by SIO                                      | STEFLUX                                                                       | SI events by SIO                                            | STEFLUX                                                                       |  |
| 1 day                                                        | 117                                                   | 25 (22%)                                                                      | 217                                                         | 45 (21%)                                                                      |  |
| 2 days                                                       | 41                                                    | 17 (42%)                                                                      | 36                                                          | 11 (31%)                                                                      |  |
| 3 days                                                       | 20                                                    | 4 (20%)                                                                       | 28                                                          | 11 (39%)                                                                      |  |
| $\geq$ 4 days                                                | 25                                                    | 9 (36%)                                                                       | 18                                                          | 6 (33%)                                                                       |  |
| Total                                                        | 203                                                   | 55 (27%)                                                                      | 299                                                         | 73 (24%)                                                                      |  |
|                                                              | NCO-P                                                 |                                                                               | Mt. Cimone                                                  |                                                                               |  |
| (b)                                                   | NCO-P                                                 |                                                                               | Mt. Cimone                                                  |                                                                               |  |
| (b)
SI event duration                                     | NCO-P
SI events by STEFLUX                         | SIO                                                                           | Mt. Cimone
SI events by STEFLUX                          | SIO                                                                           |  |
| (b)
SI event duration
1 day                     | NCO-P
SI events by STEFLUX
55                   | SIO
7 (13%)                                                                | Mt. Cimone
SI events by STEFLUX
100                   | SIO
15 (15%)                                                               |  |
| (b)
SI event duration
1 day
2 days                  | NCO-P
SI events by STEFLUX
55
36             | SIO
7 (13%)
15 (42%)                                                    | Mt. Cimone
SI events by STEFLUX
100
73             | SIO
15 (15%)
23 (31%)                                                   |  |
| (b)
SI event duration
1 day
2 days
3 days | NCO-P
SI events by STEFLUX
55
36
22       | SIO
7 (13%)
15 (42%)
7 (32%)                                         | Mt. Cimone
SI events by STEFLUX
100
73
28       | SIO
15 (15%)
23 (31%)
5 (18%)                                        |  |
| (b)
SI event duration 1 day 2 days 3 days ≥4 days         | NCO-P
SI events by STEFLUX
55
36
22
42 | SIO           7 (13%)           15 (42%)           7 (32%)           10 (24%) | Mt. Cimone
SI events by STEFLUX
100
73
28
36 | SIO           15 (15%)           23 (31%)           5 (18%)           9 (25%) |  |

**Table 3.** (a) "SIO vs STEFLUX": agreement between STEFLUX and the measured SI events (SIO), and (b) "STEFLUX vs SIO": agreement between the measured and the modeled (by using STEFLUX) SI events, as a function of the different length of the SI events.